# From Pagoda to Pavilion: The Transition of Spatial Logic and Visual Experience of Multi-Story Buddhist Buildings in Medieval China

**Yifeng Xie**

Yuelu Academy, Hunan University, Changsha 410082, China; xieyifeng@126.com

**Abstract:** Pagodas and pavilions (*ge* 閣) are the most popular and representative multi-story buildings since Buddhism was introduced to China. While providing visitors with a new visual experience, they have also largely reshaped the urban space and skyline in medieval China. The former originated from India and Central Asia and was transformed in China, developing a unique style; The latter originated more from the creation of Chinese architects and became a model of typical Chinese-style Buddhist architecture. Briefly, the pagoda matured earlier than the pavilion, and continuously developed while maintaining its basic style; the pavilion-style Buddhist architecture gradually developed later and finally matured after the Tang and Song dynasties (618–1276), partially presenting a different spatial logic from the pagoda, and bringing a new visual experience. In my opinion, although the pavilion may not necessarily be as large as the pagoda in terms of volume and absolute height, it can provide believers with greater visual impact in the internal space for worship, due to the cross-story giant Buddhist statues; the closer integration of Buddha statues and architecture makes it replace or share the core position of the pagoda in some monasteries and even become the visual center of the entire religious space. Due to the existence of the pavilion, viewers can not only worship the Buddhist statues on a two-dimensional plane or by looking up at the statues from the bottom, but have also gained a three-dimensional perspective, to worship directly at the Buddha's shoulders, neck, and head. In the Buddhist grottoes, the layout of the early single-layer or multi-layer horizontally distribution of caves on cliff was also changed due to the excavation of the cross-layer giant statue grottoes, covered by multi-story pavilion-style buildings, providing viewers with a visual experience similar to that of the pavilions of great statues. Additionally, there is a new visual experience of worshiping the Buddha in a vertical circle, in cases such as Bamiyan and the Leshan Giant Buddha.

**Keywords:** pagoda; pavilion; spatial logic; visual experience; medieval China

## 1. Introduction

As an important part of the Buddhist landscape, stupas or pagodas play an extremely significant role in the study of architectural history in South Asia, Central Asia, East Asia (China, Japan, and the Korean Peninsula), and even Southeast Asia, and are also one of the main ancient architectural heritages in these regions. Overall, previous research on Indian-style, Gandhara-style, and Tibetan-style stupas has mainly focused on their external forms, decorative themes, and evolutionary history. Meanwhile, the exploration is more abundant on the multi-story pagodas and dense-eave pagodas in East Asia, especially in China, and also involves architectural technology, internal structure, and even the expansion of their belief space in addition to the above aspects. Compared to the widely existing pagodas in the entire Buddhist world, the Buddhist pavilion-style architecture has obvious Han Chinese characteristics. Additionally, pavilions in other parts of East Asia (such as Japan, the Korean Peninsula, etc.) are also largely influenced by the comprehensively traditional Han Chinese architecture. Therefore, we can take the pavilion as the classic multi-story

Buddhist building with the most Chinese characteristics. The pavilions defined here are limited to Buddhist space, and do not include later more generalized pavilions, such as library pavilions, small pavilions in Chinese gardens, or pavilions on top of large buildings. Compared to the considerably abundant research on Buddhist pagodas[1], perhaps due to limitations in regional scope, the number of surviving cases, and to other factors, previous and current research on pavilions is relatively limited, mainly focusing on the aspects of structure, technology and decoration. In addition to a few systematic monographs, such as *Zhongguo Gudai de Mu Louge* 中國古代的木樓閣 (*Timber Pavilions in Ancient China*) (Ma 2007), the current research mainly focuses on case studies of some important buildings (such as the Guanyin Pavilion at Dule Monastery in Ji County)[2], and involves comparative analysis of *lou* 樓 (tower) and *ge* (pavilion).

In summary, there is still great potential for researchers to break the boundaries of architectural categories (such as pagodas, pavilions, and Buddhist grottoes) and materials (such as wood, bricks, stones, etc.), focusing on a systematic and detailed analysis of the spatial logic and visual experience of pagodas and pavilions in ancient China, and citing the evolution of grottoes as a reference. It should be noted that the focus of this research is not primarily on the structural analysis and technical comparison between pagodas and pavilions in the framework of the discipline of architectural history, nor on the stylistic analysis and decoration comparison of art history. Instead, this paper attempts to explore the overall spatial logic and intuitive visual experience from the perspective of designers and observers, paying more attention to their inherent differences and tensions. Briefly, the pagoda achieved earlier development of these two kinds of buildings, and was continuously updated, while continuing its overall style; the pavilion-style Buddhist architecture has gradually developed and matured since the Tang and Song dynasties (618–1276), partially presenting a different spatial logic from the pagoda, bringing a new visual experience. Both have own transition clues and transformation tracks, which are not a substitutive transition from A to B, but to some extent reflect the transformation of spatial logic and visual experience of Buddhist multi-story buildings in medieval China.[3] To provide a clearer description of the different visual experiences between pagodas and pavilions, I prefer to introduce the following two key concepts. The first is the "planar visual logic", which means viewing and worshipping Buddha statues at a relatively fixed-elevation angle on a plane; the second one is the "three-dimensional perspective", which means viewing and worshipping Buddha statues from different heights, to obtain a different elevation or even horizontal angles.

## 2. The Continuation and Transition of Planar Visual Logic in Various Buddhist Pagodas and Grottoes in India, Central Asia, and China

From ancient India of the Maurya Dynasty, to Central Asia during the Kushan Empire, and to China during the Wei, Jin, and Northern and Southern Dynasties (220–589), Buddhist stupas (pagodas) presented a trend from single-story stupas to the stupas with a multi-layer base and vertical extension after entering Central Asia, and, combined with the concept of Chinese traditional *lou*-style buildings, formed a multi-story pagoda with Chinese and even East Asian characteristics. However, due to the unchanging way of worshipping the Buddha around the pagoda, the basic visual logic of the aforementioned-pagoda also presented a typical planar surrounding logic in response to the needs of this function. Although the number of stories of the pagoda varies, it was still a vertical superposition of single-layer or multi-layer planar surrounding visual logic, each layer forming a relatively independent image program, and there was no cross-layer visual spectacle composed of giant Buddha statues. In other words, whether in India, Central Asia, China, or even the East Asian world, early stupas or pagodas did not receive more attention due to their extraordinary and massive statues spanning multiple layers, but their overall grandeur and nobility were emphasized, manifested as the superposition of multi-layer planar Buddhist sacred spaces.

In my opinion, a revolutionary transformation after the pagoda (originally the stupa) entered China was the expansion of its internal space. The widely popular skills of connecting wooden-structure corridors to the earthy core in Buddhist pagodas during the Northern and Southern Dynasties (architectural examples such as the pagoda in Mount Fangshan at Datong in Shanxi Province, the Siyan Pagoda in Chaoyang in Liaoning Province, the pagoda of Yongning Monastery in Luoyang in Henan Province, the pagoda of Zhaopengcheng Buddhist Monastery and the pagoda of Da Zhuangyan Monastery in Yecheng in Hebei Province) allowed them to obtain, to certain extent, internal space. However, the internal space of these pagodas was still quite limited.[4] What is somewhat groundbreaking is that the existence of this corridor provides people who worship at the pagoda story-by-story with a new perspective for viewing from the inside to the outside, although the visual logic of each story is still independent and vertically superimposed in a planar manner. However, during the process of climbing and overlooking, those who worship at the pagoda have gained a different visual experience from before, which is no longer limited to a single layer of flat space; they can gradually enter the upper level of flat space through stairs, experiencing the differences in visual perception brought about by the changes in three-dimensional height. In ancient China, although these multi-storied timber buildings had already existed during the Han Dynasty (BCE 202–CE 220) in the form of watchtowers, as evidenced by plenty of archaeological findings, the visual experience brought by such multi-story buildings was limited regarding personal space and limited to particular groups, like sentinels. For rulers, officials and religious groups, this visual experience was still very attractive and impactful, and was also related to the privilege of owning these buildings during the Northern and Southern Dynasties (420–589). The rulers of empire (such as Empress Ling (?–528) in Northern Wei (386–534)) attempted to monopolize this extraordinary visual experience (Yang 2018, p. 13); some officials, such as Yang Xuanzhi (active in the early 6th century), the author of *Luoyang Qielan Ji* 洛陽伽藍記, a detailed record of the Buddhist monasteries in Luoyang, and his colleague Hu Xiaoshi (active in the early 6th century) (Yang 2018, p. 13), and later nominees for the imperial examination (after the middle period of Tang, the imperial examination system flourished, and "leaving an inscription on the Wild Goose Pagoda" became a trend) attempted to share this highly unusual visual experience in high-rise buildings, located in the planarized urban layout of medieval China. It should be noted that this visual experience was still viewed from the inside to the outside in the case of external landscapes of various heights, to obtain different angles (a visual experience of three dimensions), rather than from the outside to the inside, as in the case of the internal space of the building, to view internal Buddhist statues at different heights; this did not change the planar visual logic of early pagodas. Even the pure wooden-structure pagodas (such as the Yingxian Timber Pagoda) (Liang 2007, pp. 1–118; Chen 2001), or brick pagodas with center chambers on each story obtained larger internal space due to the cancellation of the core tower entity; with the development of construction later, the stories remained independent from one another. The layout and design were still based on the unit of the story, achieving the expansion of the belief space. Due to the presence of the mezzanine level (*pingzuo* 平座), the stories with Buddhist statues are more isolated—even during the process of ascending stairs, it is impossible to see Buddhist statues arranged in different stories at the same time. In terms of Buddhist statues, due to the height limitation of each story of the pagoda, the height of the statue is still limited to the same story height. In some cases, the main Buddha statue located in the center of the pagoda may be able to slightly encroach on the upper mezzanine level through the extended space (*zaojing* 藻井, a caisson ceiling) above, but it also fails to fundamentally penetrate the cover between the stories and to break through the visual logic of the plane surrounding each story (see Figure 1). In addition, the extensive use of precious metal materials such as gold and bronze in early Buddhist statues, as well as the position on chariots during the Statues Parade, prevented them from exceeding the general scale of the one-*zhang* 丈 and six-*chi* 尺 statue (the standard height of Buddha statues in many Buddhist texts), resulting in the emergence of large-scale single giant statues. Even the un-

paralleled imperial giant structures like the pagoda of Yongning Monastery in Luoyang, as well as the Buddhist Hall behind this huge construction, have not appeared as giant statues either in the literature records (which mainly refer to the item in the Yongning Monastery in *Luoyang Qielan Ji*) or in the archaeological evidence (Buddhist statues and fragments unearthed from Yongning Monastery) (Yang 2018, p. 12; Zhongguo Shehui Kexueyuan Kaogu Yanjiusuo 1996), although there are also a few statues in the existing Great Buddha Hall that exceed the height limit of one *zhang* and six *chi*. Due to the limitations of a single-story structure, Buddhist monasteries generally still cannot accommodate giant statues of tens-of-meters high.

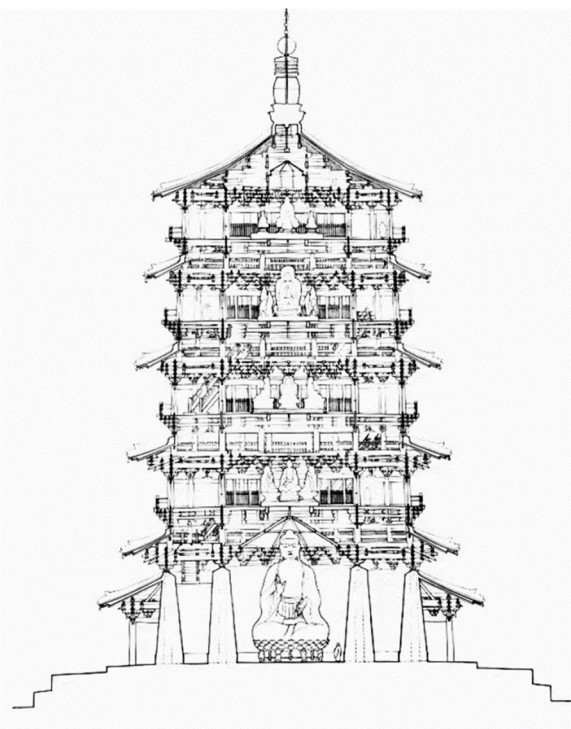

**Figure 1.** Line drawing of Yingxian Timber Pagoda. (See Liang 2007, p. 36, Figure D-1).

The form of early grottoes also followed the same logic as pagodas, with caves occupying the space on the cliff surface in a horizontal expansion manner—either horizontally located on the same level, or in a multi-layer parallel-distribution pattern, due to the length of the cliff surface. In the other words, there were few vertical expansion structures spanning multiple layers and large caves occupying multiple layers of the cliff surface in these grottoes. For instance, the Mogao Grottoes of Dunhuang were based on the "original caves" (or the so-called "three caves in Northern Liang Kingdom (397–460)"), which were the earliest constructed, and then extended horizontally to both sides and developed a few parallel caves (Wu 2022, pp. 81, 83). Even the small number of vertically distributed caves in the early Dunhuang grottoes did not make a breakthrough in terms of capacity, maintaining a spatial scale roughly equivalent to the caves on the same layer of the cliff.

If the linear horizontal extension mentioned above mainly exists in the cliff surface of the grottoes, the spatial logic of horizontal encirclement is mainly reflected in the interior of the grottoes. In the "Caitya or Chaitya" in India, the spatial pattern of horizontal worship around the stupa is applied to the interior space of caves, presenting the same visual logic (Li 2014, pp. 3–20), although the "central pagoda pillar cave" in the grottoes of the Han Chinese exhibits a multi-story composite feature on its central pillar and forms a corresponding multi-layer pattern with the four walls of the caves; fundamentally speaking, it is still arranged in a horizontal layered manner, without changing very much the basic visual logic of early Buddhist pagodas and grottoes.

Additionally, it is necessary to focus on analyzing a so-called "exception", which are the earlier giant Buddha caves among the grottoes from the Northern Dynasty (439–581), especially the 16th to 20th Caves of the Yungang Grottoes. However, the main deity of such caves may not be on the same horizontal line as the surrounding attendants (attendant Bodhisattvas), presenting a certain degree of dislocation; according to Peng Minghao's research, the existence of this phenomenon is due to the layered construction of the main Buddha statue (Peng 2017, pp. 66–254) (See Figure 2). Fundamentally, although the caves in this group have huge capacities, their layout still extends horizontally. Although the lighting holes in the upper part of each cave seem to present a visual perception as in the second layer, the holes still fail to change the basic visual logic of these caves in Yungang. In other words, these huge caves are only distributed on one larger level, rather than on the real two levels of the surface of cliff.

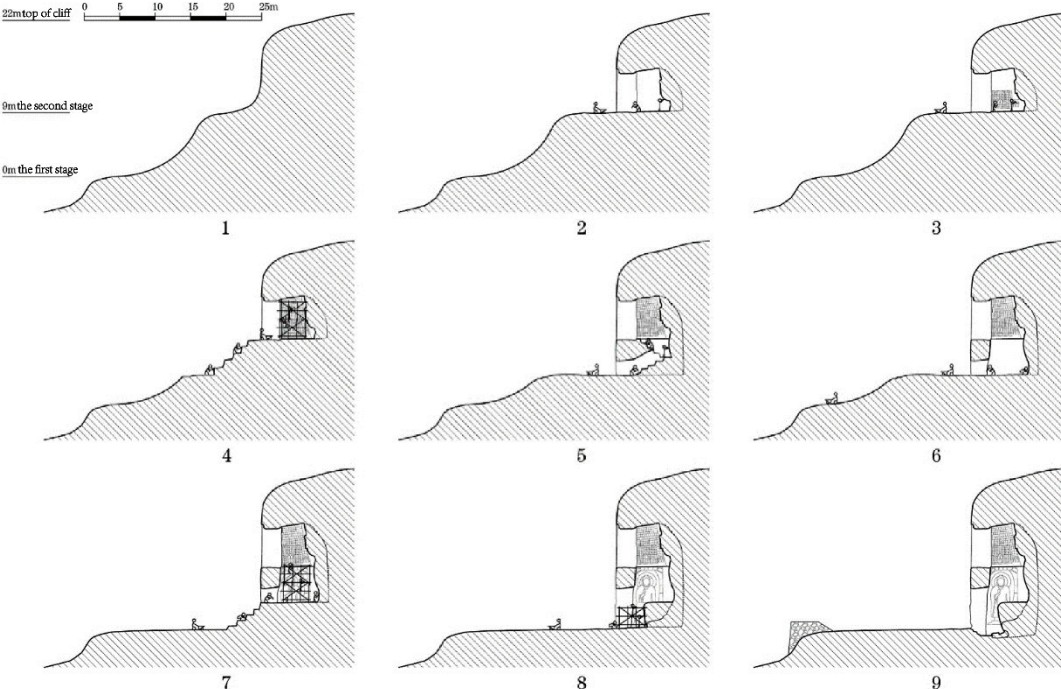

**Figure 2.** Schematic diagram of excavation process of Cave 17 at Yungang Grottoes. (See Peng 2017, p. 85, Figure 4.27). Numbers 1–9 represent the nine stages of cave excavation.

Based on the records in the *Weishu Shilao Zhi* 魏書·釋老志, a history of Buddhism and Daoism in Northern Wei, five heavy bronze Buddha statues (50,000 *jin* 斤, or roughly 24,000 kg corresponding to each one) are enshrined in the core pagoda of the monastery, corresponding to the five emperors of Northern Wei in the Wuji Monastery (a large imperial monastery with five layers of pagodas) in Pingcheng, the capital city of Northern Wei, now Datong in Shanxi Province (Wei 1974, p. 3036).

Considering the weight-bearing capacity of each story and the capacity of the space inside the pagoda, these bronze statues were likely arranged sequentially from bottom to top or from top to bottom, and each statue in one story corresponds to one emperor. In comparison, the "Five Caves Built by Tanyao" flatten the vertical multi-story structure to a horizontal straight line, with the west as the most prestigious position, distributing all five caves in a horizontal line on the surface of the cliff.

According to Peng Minghao's research on the construction project of Yungang Grottoes, the creators of the Big Buddha Caves, represented by Cave 16 to 20, used existing cliff surfaces and two-story platforms to manually treat the natural cliff surface (so-called "mountain cutting"), which was layered from top to bottom, to make it more perpendicular to the ground. If so, in a position flush with the lower edge of the lighting hole on the front

surface of the Big Buddha Caves, there should be the working platform for carving the Buddha's head, shoulders, and chest. Due to the time-consuming and laborious process of carving the Buddha's head, craftsmen may have carved on this working platform for a longer period, which explains well the fact that several statues in today's cave that appear to be far from the ground inside the cave have lower edges that are flush with the platform of this previous project (used for carving the Buddha's head, shoulders, and chest) (Peng 2017, pp. 44–65). From this perspective, the craftsmen who participated in the carving of the statues or the visitors who entered the caves at this time should be able to view the upper half of the Buddha statue from a close viewpoint, obtaining a new visual experience that is different from the past. However, the visual experience above neither exists permanently nor corresponds to the creator's purpose, but rather is a phased result of the process of advancing the working platforms layer by layer. With the downward extension of the statue project, the above-mentioned working platform that was flush with the lower edge of the lighting hole eventually disappeared, and the access to this phased platform was completely blocked. After the completion of subsequent projects, viewers entering the cave can only look up and worship the statue from the bottom of the cave or view the complete statue from outside, and no longer obtain the special viewpoint and visual experience that craftsmen and supervisors once had during the construction process.

## 3. The Possibility of Layered Viewing and the Structural Characteristics and Visual Experience of the Pavilion of the Giant Buddhist Statue

To continue the previous discussion, although the layered-construction technology of the giant statues in Yungang Grottoes has not fundamentally changed the visual layout of its horizontal extension, it suggests a possibility of layered viewing. If this visual experience no longer disappears due to the disappearance of the working platform, but can be preserved through some external structure, this fundamentally means a breakthrough in the way of viewing and the visual logic of Buddhist statues. Therefore, the key to transforming this temporary visual experience into a completely new viewing experience and visual logic is the establishment of the architectural form that supports this viewing experience. Generally, the key structure that can provide this visual experience is comparable to the cross-story hollow structure that exists in today's large shopping malls. In medieval China, the main one capable of fulfilling this functional demand was the pavilion (*ge*). From the exterior facade, the pavilion is also a typical multi-story structure, like the pagoda. However, unlike the independent structure on each story in pagodas, the interactivity of the spaces on each story within the pavilion has greatly increased, especially with the emergence of cross-story core spaces that can accommodate giant Buddha statues. The practice of establishing multi-story pavilions in Buddhist monasteries began in the late Northern and Southern Dynasties (420–589). On the one hand, there are similarities in the special structure between Buddhist monasteries and imperial or aristocratic high-ranking buildings, since many imperial and aristocratic residences were donated to Buddhism as monasteries at that time; on the other hand, the structure was also related to the way of setting up Buddhist statues (Fu 2009, p. 511). In the *Chang'an Zhi* 長安志, written by Song Minqiu (1019–1079), the basic structure of the Buddha pavilion at Baocha Monastery in Chang'an was recorded during the Northern Wei Dynasty. The description of "erecting pillars on four sides, forming an elevated space in the middle, and establishing a two-story pavilion" 四面立柱，當中虛構，起兩層閣 (Song 1990, p. 114) clearly indicates the existence of a cross-story space within the pavilion that can accommodate large Buddha statues, surrounded by pillars on four sides. Therefore, although this type of pavilion-style building often lacks the overall height and capacity of those super high-rise pagodas, it has an internal space that is more integrated and not completely horizontally separated.

By the time of the Sui and Tang dynasties (581–907), there were many historical records of worshipping giant Maitreya statues in pavilions. According to the section of "*Quchi fang* 曲池坊" in *Chang'an Zhi*, Jianfu monastery "was established by the Princess of Xincheng in the third year of the Longshuo period (663). Its location was originally Tianbao Monastery in



Sui Dynasty (581–618). Inside the monastery is the Maitreya Pavilion built in Sui, which is 150 *chi* high. 龍朔三年爲新城公主所立，其地本隋天寳寺，寺内隋彌勒閣，崇一百五十尺" According to the standard of Sui, one *chi* is equivalent to 29.6 cm (Guo 2008, p. 191), and the height of the Maitreya Pavilion can reach 44.4 m. On the column of Dhāraṇī Sutra built by a Buddhist nun in the Longhua Monastery of Tang, it is also recorded that there is a Maitreya Pavilion and large Buddha statues in the pavilion of the Longhua Nuns' Monastery in Qujiang, Chang'an (Lu 1985, p. 321). The biography of Faxing in *Song Gaoseng Zhuan* 宋高僧傳, a collective biography of eminent monks from Emperor Gaozong' reign in the period of Tang (649–683) to Early Northern Song (960–1127), also records that there was a Maitreya Pavilion of three stories and a width of seven *jian* 間[5] in the Foguang Monastery of Mount Wutai, which was 95 *chi* (about 28.5 m) high (Zanning 1987, p. 690). To sum up, during the Sui and Tang dynasties, the Pavilions of Giant Statues, which were 30 to 40 m high, were not uncommon in large monasteries around Chang'an and Wutai.

Since the pavilion can be accessed and its central space can be connected to various stories, visitors have a completely different visual experience. Their way of viewing giant Buddha statues is no longer limited to looking up from the front or from the bottom, but they are now able to see the upper part of the Buddha's body, shoulders, and head through climbing the pavilion, obtaining a visual experience that was not previously available in pagodas or grottoes in the early period. While feeling the majesty of the colossal statue, viewers can also closely observe its detailed features from different angles, and even gaze directly and horizontally into the eyes of the Buddha statue. For instance, the Japanese monk Ennin (793–864) came to the Tang Empire to search for Buddhist sutras and doctrines. When he arrived at Kaiyuan Monastery in Taiyuan, he at once "climbed onto a pavilion to observe (the Buddha). Inside the pavilion, it is a statue of Maitreya Buddha, cast in iron and painted in gold. The Buddha's body was over three *zhang* long and sat on a throne 上閣觀望。閣內有彌勒佛像，以鐵鑄造，上金色。佛身三丈餘，坐寳座上" (Ennin 2019, p. 312). Based on this record, it can be clearly stated that although the size of the Maitreya seated statue in this pavilion is not very large, it has significantly exceeded the conventional height of one *zhang* and eight *chi*, reaching about ten meters. The method of iron casting must have been very popular in Tang. According to Zheng Yan's research, the so-called "iron cassock" located at Lingyan Monastery in Jinan, Shandong Province, is actually a partial remnant of clothes from a statue of warrior attendants. Based on the height of the fragments of the cast iron statue, it can be inferred that its original height was about 7 m. The originally complete statue was destroyed during the extermination of Buddhism in the period of Huichang (841–846) (Zheng 2006, pp. 206–14; Zheng 2022, pp. 30–36). According to hierarchical differences, as the main deity, Vairocana Buddha should be bigger, close to the height of the Maitreya statue recorded by Ennin. According to *Xijin Zhi* 析津志, cited by *Shuntianfu Zhi* 順天府志, a significant local gazetteer of Beijing in early Ming (1368–1644), Minzhong Monastery "is located in the south of the old city, with a giant pavilion dedicated to the statue of Guanyin in white, which is more than twenty *zhang* high. The head of the statue can only be seen on the third story of the pavilion 在舊城之南，有傑閣奉白衣觀音大像，高二十餘丈。閣三層始見其首。" (Xie 2017, p. 15) If this record is true, the height of the statue of Guanyin in white can reach over sixty meters. More importantly, for these pavilions which have disappeared, the visual experience of only seeing the head of the Bodhisattva when the viewers climbed to the third story is not something we can imagine today, but was faithfully recorded by the viewers at that time.

For circular Buddha statues, this design can also facilitate the viewer in circling around the back of the Buddha statue, which is also a visual experience that cannot usually be provided by Buddha statues attached to the central column of pagodas or located near the back wall of Buddhist halls. The Guanyin (Avalokiteśvara) Pavilion of Dule Monastery is a representative example (see Figure 3). On the one hand, it is comparable to the Yungang Caves 16 to 20 in the two-story design and basic spatial composition created by the Guanyin statue inside, and even to the lighting setting of the two-story open windows in the Guanyin Pavilion. On the other hand, for visitors who visit this pavilion, viewing the

chest, abdomen, head, and neck of the Guanyin statue from a close distance through the inner corridor on the mezzanine level or second story is no longer a privilege for craftsmen in the process of grotto construction, nor is it a phased experience during the construction process. Instead, it has become a part of its design philosophy and visual logic. The viewing of the back of the Bodhisattva statue is a visual experience that cave craftsmen have almost never had. The other typical case is the Dabei Pavilion of Longxing Monastery in Zhengding County, Hebei Province (see Figure 4). People can also see the head, chest and abdomen of the giant Guanyin statue horizontally by climbing on the pavilion. Unfortunately, since the current Dabei Pavilion is not the original structure of Northern Song, and is not from the same time as the Guanyin statue inside the pavilion, it is still not possible to accurately analyze the visual experience of the Song people who climbed onto the pavilion and observed this statue, as in the Dule Monastery.

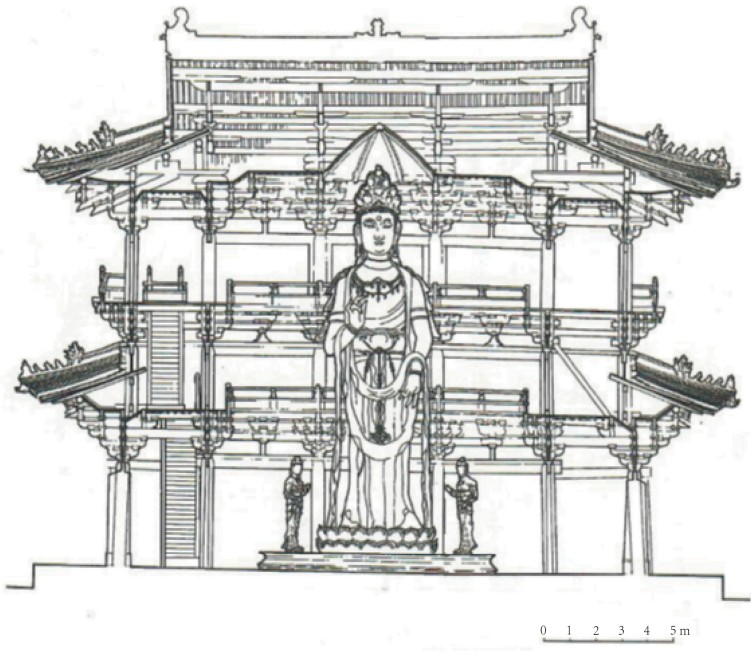

0  1  2  3  4  5 m

**Figure 3.** Vertical section of the Guanyin Pavilion of Dule Monastery. (See Guo 2009, p. 290, Figures 6–29).

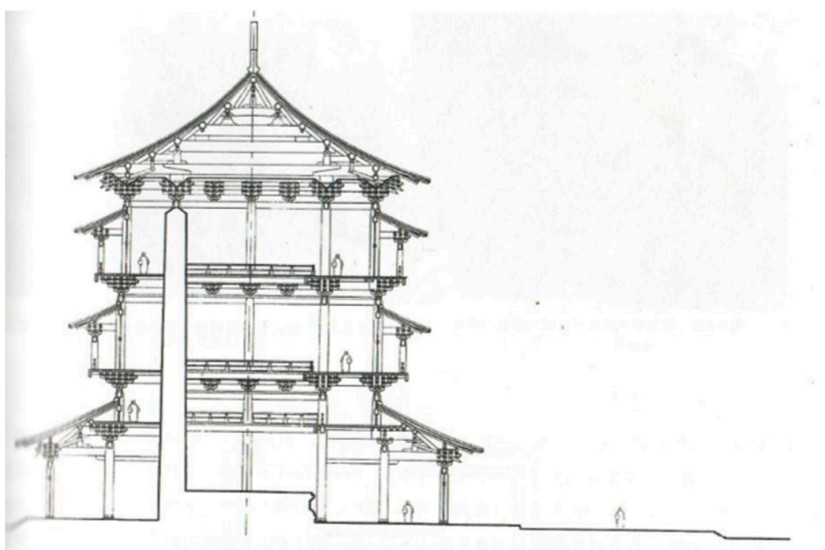

**Figure 4.** Section of restoration of the Dabei Pavilion in Longxing Monastery (Guo 2009, p. 373, Figures 6–214).

According to the stele in 963, there is a copper statue of Bodhisattva inside the Dabei Pavilion, which is seven *zhang* and three *chi* high. In 1316, Zhao Mengfu (1254–1322) described this huge building in the so-called "Danba Stele": "Its statue is 73 *chi* high, and a large pavilion of three stories was built to cover it. On the side are two magnificent multi-story buildings like wings, which are unparalleled in the world 其像高七十三尺，建大閣三重以覆之。旁翼之以兩樓，壯麗奇偉，世未有也." (Wu 2019) In 1438, the other stele recorded the height of the pavilion as 13 *zhang*. According to the records in this stele and my interpretation, there are auxiliary buildings on both sides of the pavilion, each measuring seven *zhang* and three *chi* high, which is exactly equal to the height of the giant Bodhisattva statue. Chinese architectural historians speculate that the Dabei Pavilion in Song was a building with a width of seven *jian* and a depth of six *jian*, based on the existing ruins of the pillar foundations and the traces of the platform in Song. Based on the functional and structural needs, the height of the pavilion needs to be determined based on the size of the existing copper Buddha. The measured height of the copper Buddha statue is 21.3 m, the base is 2.35 m, and the total height is 23.65 m. According to these data, the Dabei Pavilion is designed as a three-story pavilion (with four eaves and three floors as the main body, which is not fundamentally contradictory to Zhao Mengfu's description) in a reconstructed imaginary drawing. In the center of this building, there is a high space connecting four stories, surrounded by rectangle internal corridors, and the roof type of the top of the pavilion is a hip–gable roof. The total design height of the pavilion is 37 m, and with the addition of roofs it roughly matches the record which says that "the pavilion is thirteen *zhang* high 閣高十三丈", in the historical resources (Guo 2009, p. 371). Based on photos taken from the early 20th century to the 1920s and 1930s, the main structure of the pavilion was divided into three stories, with four or five eaves on the exterior.[6] The first story had single eaves, the second story had single or double eaves, and the third story had double eaves. There was an obvious mezzanine level between each main story. From the existing buildings that are said to have been restored according to the structure in Song, the front and sides of the Guanyin statue can be viewed from the east, west, and south. Due to the presence of a back supporting wall, it is not yet possible to see its back. The inner corridors are divided into three levels, corresponding to the lower part of the mezzanine level between the first and second main stories, the upper part of the same mezzanine level and the third main story, which are flush with the legs, waist and head of the Guanyin statue. Standing on the third-level corridor, visitors can look at the head of the Bodhisattva statue and observe many details.

The Mahayana Pavilion of Puning Monastery, built in 1755, is also a three-story giant structure (with varying numbers of eaves in different directions), with a total height of 36.65 m (see Figure 5). The interior hall accommodates a giant timber statue of Guanyin that exceeds 27 m. Standing on the second and third level of internal corridors in the pavilion, visitors can almost reach and touch the open arms of the Thousand Hands and Thousand Eyes Guanyin, to closely experience the charm and carefully observe details of this statue. Because current viewers can only look up from the bottom, their visual perception of this Guanyin statue is more of a sense of oppression and authority from above. However, for the privileged group who had the opportunity to climb onto the second and third story of this pavilion at that time, the huge Guanyin statue was not only visible, but also within reach, offering a completely different visual experience. It is very interesting that this wooden carving of the Guanyin statue is made of numerous timbers, with a hollow interior divided into three layers by horizontal structures and supported by a center pillar that penetrated through the top and bottom. In the other words, there are also layered hidden spaces inside this Bodhisattva statue. Although its main structure appears to be integrated as a whole, in essence it can be seen as an unconventional multi-story building. The construction method of its main statue also testified to the technical feasibility for the giant statue in the Heavenly Hall of Empress Wu (614–705, r. 690–705), which we will discuss next.

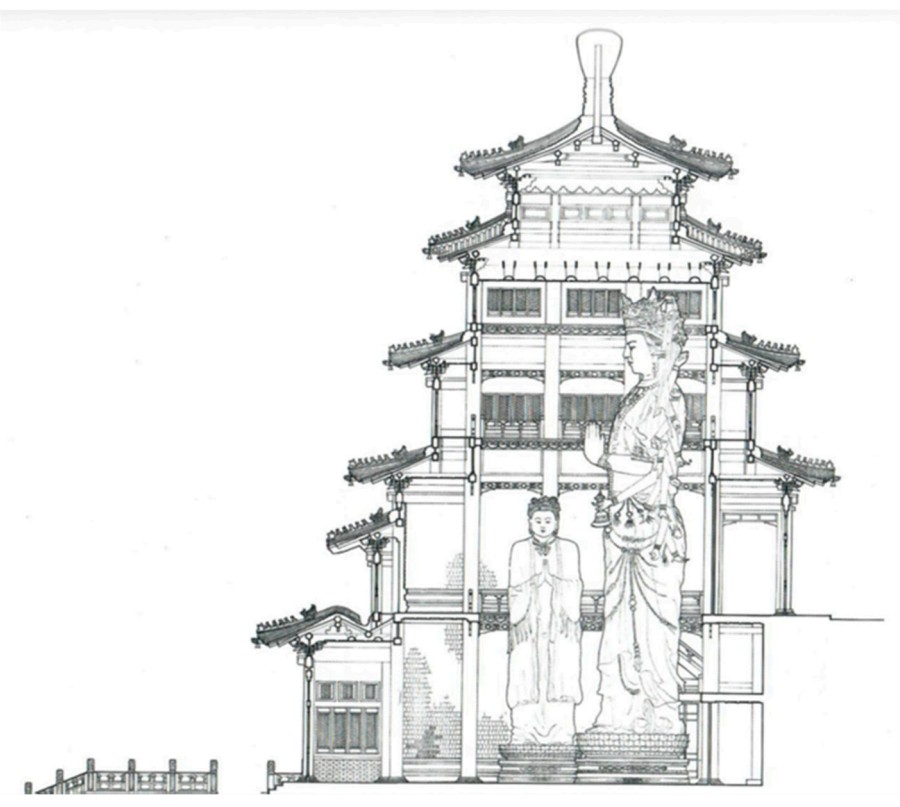

**Figure 5.** Cross section of the Mahayana Pavilion in Puning Monastery (Yang and Zhu 2019, p. 137).

The above-mentioned three examples are all Guanyin statues after the Liao Dynasty (907–1125) and Song Dynasty. However, the motif of this type of cross-story giant statue is clearly not limited to Guanyin in cases nowadays. It is like the Wanfu Pavilion in the Yonghe Palace (the Lama Temple in Beijing), the former residence of Emperor Yongzheng (1678–1735, r. 1722–1735), which has three stories (with two stories and three eaves on the exterior and one underground dark story on the interior). In the center hall of this pavilion, there is a giant Maitreya statue that is up to 26 m high, 18 m above ground, and 8 m underground, carved from a whole piece of sandalwood. Unlike the previous pavilions, due to the height limitation of the Wanfu Pavilion itself (which is much lower than the Maitreya statue on the interior), a considerable portion of the legs of Buddha statue extends into the underground space, and its chest is flush with the floor of the second story of the pavilion (due to the presence of underground space, it is the third story from the inside). Therefore, although the main deity of the Wanfu Pavilion in the Yonghe Palace is Maitreya rather than Guanyin, and some of it is located underground, its basic visual logic remains consistent with the aforementioned Guanyin Pavilion and Dabei Pavilion.

Beyond these existing architectural cases, the most extreme masterpiece is the Heavenly Hall built in Luoyang, the holy capital of the empire during the Wu Zhou period (690–705). According to relevant historical records and Luo Shiping's research, the Heavenly Hall (*tiantang* 天堂) was in fact the Buddha Hall, which accommodated a giant statue of Maitreya (Liu 1975, p. 865; Du 1984, p. 1228; Luo 2016, pp. 30–35). But to define it as pagoda-style architecture is at least inaccurate. The internal structure of the Heavenly Hall was relatively close to the pavilion, except that its plane was circular rather than the rectangle or octagonal shape commonly seen in existing pavilions. The internal structure of the network distribution of multi-circle pillars allowed it to meet the needs of those climbing onto the pavilion and to form a hollow structure connecting multiple stories, accommodating cross-story giant statues. Compared to the Guanyin Pavilion of Dule Monastery, which has fewer stories, the Heavenly Hall, with more stories (the exterior appears to have five stories, with four additional mezzanine levels, and is composed of nine structural stories

in total), allowed visitors to observe the Buddha's feet, body, and head in proximity, providing rich perspectives from all directions and elevations at more different heights. If in the Dabei Pavilion of Longxing Monastery in Zhengding County the inner cloisters were set at the lower and upper parts of each mezzanine level, the viewing positions at different heights in the Heavenly Hall are set at eight different layers. From the actual measurements provided by the archaeological report, the network of pillars of the Heavenly Hall is concentric in shape, with two circles of stone column foundations arranged with an inner circle of twelve and an outer circle of twenty (see Figure 6). The central part of the building has a super-large foundation composed of several large stone slabs, indicating the possibility of the existence of a huge central pillar (Luoyangshi Wenwu Kaogu Yanjiuyuan 2016, pp. 26–32). However, in my opinion, if this site is indeed the site of the Heavenly Hall, there is definitely a giant Maitreya Buddha statue that spans multiple stories in it, according to historical records (Liu 1975, p. 865; Du 1984, p. 1228; Luo 2016, pp. 30–35). Nevertheless, the distance between the inner pillar network and the central pillar, at most ten meters, may not be enough to fully accommodate a giant statue that is more than 100 m high (the speculated height of Maitreya's statue is detailed later). Even if it can accommodate it, it would still cause the giant statue to be obstructed by the inner pillar network, and the height of the statue does not match the design height of the pavilion. Referring to the actual Guanyin statue in the Mahayana Pavilion of Puning Monastery, a dramatic but reasonable assumption is that this central pillar may not be the core building component that reaches the top of the Heavenly Hall, but rather a supporting central pillar wrapped inside a lightweight-design (*jiazhu* 夾紵) Buddha statue. In other words, what it wanted to support was not the Buddha Pavilion, but the giant statue. If the size of this Maitreya statue was indeed as huge as described in historical records (Liu 1975, p. 865; Zhang 1979, p. 115), it is highly likely that its interior also adopts a layered hollow structure like the Guanyin statue of Puning Monastery, with a central timber pillar connecting the horizontal layers, to support the whole structure. Due to the existence of networks composed by at least two circles of pillars in the site of the Heavenly Hall, these pillars can form a circular inner corridor and ensure that the central area is connected from top to bottom. Therefore, upon climbing onto the statue and looking inward from the internal corridors, visitors can observe and worship the feet, legs, abdomen, chest, and even head of Maitreya's statue layer by layer, from a horizontal position. As for whether the interior of the hollow Buddha statue retains a certain layered structure, and can even be climbed onto step by step, providing a visual experience like the interior of the Statue of Liberty, this is uncertain, due to the lack of historical and archaeological records. As shown in the interpretation by Luo Shiping of the giant statue in the Heavenly Hall, this Maitreya statue was in Luoyang, the capital city of the empire and played an exemplary role in the Tang Dynasty. After its completion, it strongly promoted the trend of carving Maitreya statues in capitals and even in various prefectures (such as Dunhuang and Jia Prefecture) (Luo 2016, p. 29).

Later, the Heavenly Hall was burned down; the statue of Maitreya enshrined inside was not completely destroyed, but was restored by Emperor Zhongzong (656–710, r. 683–684, 705–710) of Tang. According to *Sui Tang Jiahua* 隋唐嘉話, a book edited by Liu Su (active in the period of Emperor Xuanzong [685–762, r. 712–756]), in order to record many stories of figures in Sui (581–618) and early Tang, "During the reign of Emperor Zhongzong, in order to fulfill the wishes of Empress Wu, he cut off the Buddha (Maitreya) statue, shortened it, and established a new pavilion in the Shengshan Monastery to accommodate it 至中宗欲成武后志，乃斫像令短，建聖善寺閣以居之."[7] (Liu 1979, p. 38). This record indicates that the design height of heaven should match the giant Maitreya statue built earlier. Therefore, the newly built pavilion of Shengshan Monastery in the period of Emperor Zhongzong could only be accommodated by truncating the Buddha statue, due to the pavilion's insufficient vertical height, to accommodate the originally giant statue. Regarding the height of the statue of Maitreya, Li Chuo (?–862) cited the "Record of the Great Statue of Baoci Pavilion in Shengshan Monastery" in his *Shangshu Gushi* 尚書故實: "From the top to the *yong*, it is 83 *chi*, and the mercy bead is made of silver. The hole in

the forehead (*baihao* 白毫) can accommodate items weighing eight *dan* 石 (about 635 kg) 自頂至頤八十三尺，慈珠以銀鑄成，虛中盛八石." The meaning of *yong* 顒 is quite difficult to understand here. Fortunately, the other record, *Nanbu Xinshu* 南部新書 by Qian Yi (968–1026), indicates that "the statue of the Buddha in the Baoci Pavilion of the Shengshan Monastery is 83 *chi* from the head-top to the *yi* 頤 section […]". Clearly, "Yi" means the chin, so it can be interpreted that the Buddha statue reaches 83 *chi* from the top of the head to the chin, which is 25 m according to the length of the standard *chi* in Tang. This distance can be roughly regarded as the height of the Buddha's head. From existing examples, the ratio of the head to the body of the Buddha and Bodhisattva statues in Tang ranges from 1:4 to 1:6. Even considering that the proportion of the head of the statue in the Heavenly Hall is slightly increased, due to the effect of perspective, from looking up (for example, the South Giant Statue of Cave 130 in the Mogao Grottoes of Dunhuang), it should not be lower than 1:4. Therefore, it can be inferred that the original height of the statue of Maitreya in the Heavenly Hall should be over a hundred meters, and was an unparalleled giant statue in the world at that time. The biography of Xue Huaiyi (662–695) in *Jiu Tangshu* 舊唐書 records the height of the Bright Hall of Wu Zhou Dynasty (*wuzhou mingtang* 武周明堂) as 300 *chi* (Liu 1975, p. 4742)[8], and the article by "Mingtang" in *Tongdian* 通典,edited by Du You (735–812), states: "When the Bright Hall (*mingtang*) was first built, the five-story Heavenly Hall was built behind the Bright Hall. To stand on the third story, it was already possible to overlook the Bright Hall 初為明堂，於堂後又爲天堂五級，至三級則俯視明堂矣." (Du 1984, p. 1228) Based on this description, it can be inferred that the design height of the five-story Heavenly Hall should be much higher than that of the three-story Bright Hall, at about 500 *chi*, or 150 m, which is enough to accommodate a giant statue of over 100 m. In addition, according to the following record in *Jiu Tangshu Liyi Zhi*: "At that time, (Wu) Zetian built the Heavenly Hall at the location of the original Daye Hall in Sui (581–618), to the north of the Bright Hall, to accommodate the Buddha statue, which was over a hundred *chi* high 時則天又於明堂北隋大業殿處造天堂，以安佛像，高百餘尺." (Liu 1975, p. 865) If this record was correct, the height of the Heavenly Buddha statue was only over thirty meters. In my opinion, although this information comes from official history, it appeared relatively late and was inconsistent with other data. For these reasons, I prefer not to accept these data. In addition to this overly conservative description, another overly exaggerated record is from *Chaoye Qianzai* 朝野僉載 by Zhang Zhuo (active in the period from Emperor Gaozong to Xuanzong, in Tang). According to this text, the height of the Heavenly Hall is recorded as one thousand *chi*, nearly 300 m; the height of the Buddha statue is 900 *chi*, reaching 270 m (Zhang 1979, p. 115), which is over-exaggerated and does not match the proportion of the head height of the Buddha statue recorded by Li Chuo and Qian Yi. Therefore, we also cannot accept these data. In short, both the original Heavenly Hall and the subsequent huge pavilion in the Shengshan Monastery, used to accommodate this Maitreya statue, have exemplary and benchmark significance as pavilions of giant Buddhist statues, due to their enormous size that surpasses existing examples.

We often see the words *fu* 覆 (cover) and *rong* 容 (accommodate) in historical records related to the pavilions of giant Buddhist statues. These Chinese words like *fu* and *rong* emphasize the core position of Buddha statues in the structural logic of pavilions of giant Buddha statues, and this explains the primary and secondary relationships between Buddha statues and pavilions. Firstly, it can be called covering when the giant Buddhist statue was built first and then the pavilion was constructed later. Secondly, it can be called accommodation when the Buddhist statue was the main core and the pavilion was built as the auxiliary. In other words, the design of the pavilion is subject to the internal giant statue, and its height and other parameter indicators are also set based on the size of the internal giant statue. Only when it is necessary to relocate the giant statue (such as moving the giant statue of Maitreya, originally in the Heavenly Hall, to the pavilion in the Shengshan Monastery) will there be a situation of transforming the giant statue to fit the height of the lower pavilion. Another rare possibility is that the design of the pavilion can only reach a certain limited height and number of stories, due to the type of architecture,

as in the case of the Wanfu Pavilion in the Yonghe Palace. In this situation, it is possible to search for space underground and place the bottom of the statue in a sunken space inside the pavilion, to accommodate the giant statue. In this way, higher giant statues can be placed in a relatively low space. From the overall three-dimensional design perspective, the relationship between the pavilion and its internal statues is fundamentally different from the situation in pagodas. Regardless of whether the pagoda contains Buddha statues or not, the impact of Buddha statues on its design structure can be considerably limited and will not have a fundamental impact on the shape and technical parameters of the pagoda. In this sense, the Buddha statues in the pagodas have distinct decorative and external characteristics. Even the Buddha statues in Yingxian Timber Pagoda, composed of a multi-layer mandala, still do not impact the originally designed structure of the pagoda. According to the timeline, the history of stupas or pagodas is even longer than that of Buddhist statues, and from the symbolic perspective of Li Chongfeng, "the pagoda is not only a visual symbol of Buddhist power, but also a symbol of Buddhist reverence and eternity. Moreover, it can also be seen as the embodiment of the Buddha 塔既是佛教統治的視覺標志，也是佛法尊崇和永恆的象徵。 而且，它還可以被看作佛主的 化身" (Li 2014, pp. 5–6). Therefore, pagodas can exist without relying on Buddha statues, which is the fundamental difference between them and the pavilions of giant statues.

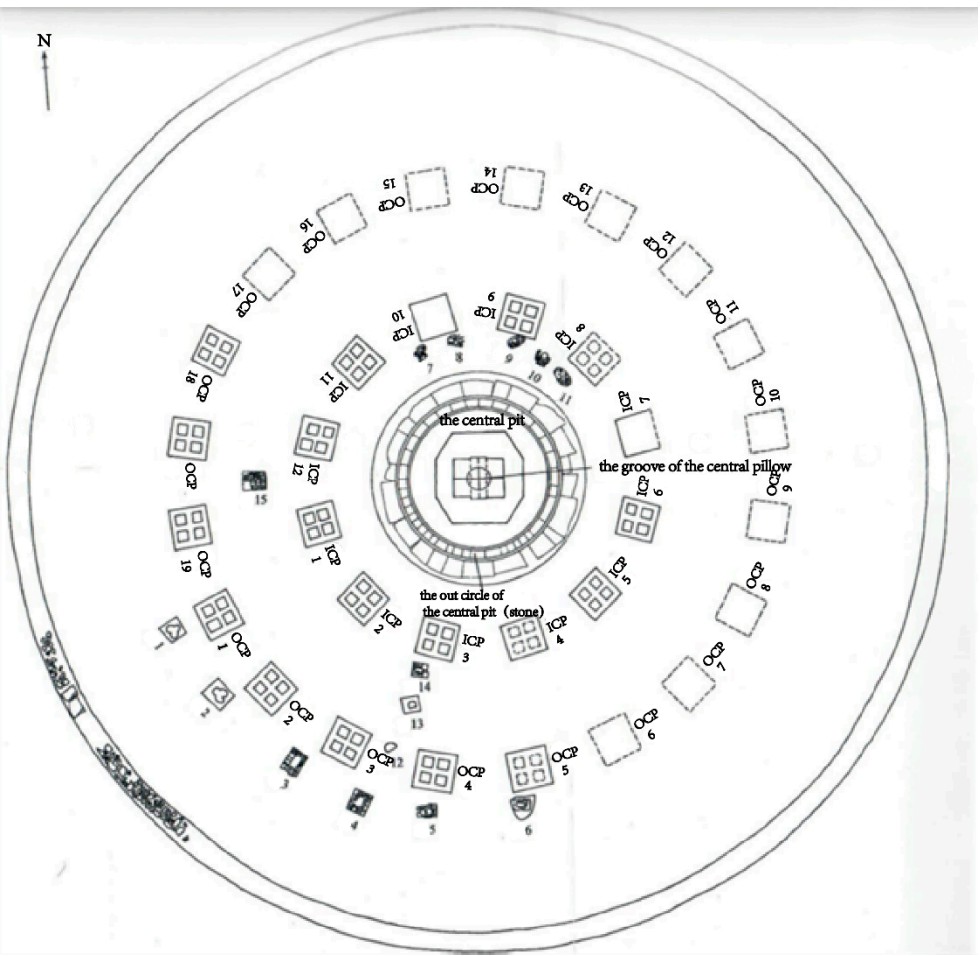

**Figure 6.** Plan of restoration of the round building (the Heavenly Hall) (see Luoyangshi Wenwu Kaogu Yanjiuyuan 2016, Figure 16). ICP: inner circle of pillars; OCP: outer circle of pillars.

The above section mainly discusses the three-dimensional structure and visual experience of the pavilions of giant statues. In terms of plane layout, combined with the descriptions in *Guanzhong Chuangli Jietan Tujing* and *Zhong Tianzhu Sheweiguo Zhiyuan Si Tujing* written by Daoxuan (596–667) and the accompanying drawings of the engraved version

in the Southern Song Dynasty (1127–1276) (Daoxuan 2018), and according to the illustrations drawn by the modern scholars of architectural history and the existing architectural examples, the main positions of the pavilion can be divided into two types: one is located on the left and right side of the main axis in the form of accessory buildings, and the other occupies the central axis as the main building.

The pavilions classified as accessory buildings generally have only two stories and are limited in height, such as the Puxian (Samantabhadra) Pavilion of Shanhua Monastery in Datong, Shanxi, and the Cishi (Maitreya) Pavilion of Longxing Monastery in Zhengding, Hebei. In the case of the Puxian Pavilion, there is a mezzanine level to structurally connect the lower and upper main story, which is made of so-called *chazhu zao* 叉柱造 (the upper pillar is inserted on the *ludou* 櫨斗 (a near-square wooden structure used for vertical support on the top of pillars in traditional Chinese architecture) of the mezzanine level and is indented inward by half the diameter of the lower pillar), and the structures of the two main stories are relatively independent. In the case of the Cishi Pavilion, it is also a two-story structure, and uses Yongding pillars that connect the upper and lower parts as a whole structure. In the front of the statue, the method of reducing pillars is used to form a larger and leaping-stories central space. Viewers can climb up the stairs at the back of the building and stand on the inner corridor of the second story, horizontally paying respects to the treasure crown of Maitreya Bodhisattva, and obtaining a visual experience like in the Guanyin Pavilion of Dule Monastery. The internal interconnected space is enclosed by four *yongding* 永定 pillars, which are integrated pillars that connect the upper and lower stories, which is different from the hexagonal structure of the Guanyin Pavilion.

The pavilion located at the core position of the central axis is even higher and can accommodate more giant statues; these become the commanding height and visual center of the entire monastery, as in the role and position of pagodas in early Buddhist monasteries, sharing this axis with them in the later period. According to the biography of Huiyun in *Song Gaoseng Zhuan* 宋高僧傳, there was a Buddhist pavilion behind the front hall of the Xiangguo Monastery in the Bian Prefecture. It was built in 745 and named *paiyun* 排雲 (Zanning 1987, p. 660). *Can Tiantai Wutaishan Ji* 參天台五臺山記, written by Jojin (1011–1081), records that Puzhao Wang Monastery in Si Prefecture also has four-story pavilion behind the Buddha Hall, called Baoyan Pavilion (Jojin 2009, p. 246). In existing buildings, the Guanyin Pavilion of Dule Monastery in Ji Prefecture and the Dabei Pavilion of Longxing Monastery in Zhengding county both worship the giant statue of Guanyin, which is located on the central axis of the monastery. These pavilions are the core buildings in the monastery, because of their remarkable ranking and height. Additionally, there are some cases which had already disappeared but were recorded in historical texts, such as the Manjusri Hall in the monasteries of Mount Wutai and Chang'an during the Daizong period (762–779), and the Dabei Pavilion seen by Khitan rulers when they entered Yanjing (nowadays Beijing). Since there is no clear record, it must not have been a symmetrical accessory building, but a core main building on the axis of the monastery. It should be noted that the appearance of the Pavilion of Great Statues located on the axis or at the geometric center of the sacred space of Buddhist monasteries is not a simple replacement for pagodas. The relation between them is not a procedure from A (pagodas) to B (pavilions), but rather coexists within the sacred space of Buddhism. In some specific cases, such high pavilions closely related to Buddhism had broken through the limitations of Buddhist space and become the center of the capital. The Bright Hall with obvious Buddhist elements; the Heavenly Hall, where the giant statue of Maitreya was located during the Wu Zhou period; the Da'an Pavilion, the main hall to for arranging Buddhist statues in Xanadu (located about 20 km northeast of Shangdu Town, Zhenglan Banner, Xilingol League, Inner Mongolia Autonomous Region); and the Central Pavilion located in the Da Tianshou Wanning Monastery in Dadu (now Beijing) during the Yuan Dynasty (1271–1368), which occupied the geometric center of the palace or capital city, had already become significantly political landscapes, with Buddhist characteristics.

### 4. Interactivity and Divergence in the Comparative Analysis of the Visual Logic of Caves of Giant Statues, Pavilions of Giant Statues and Pagodas

At almost the same time as the development of the pavilions of giant statues, caves of giant statues which occupy almost the entire cliff surface also appeared in Buddhist grottoes.[9] Judging from several cases and existing sites, the giant Buddhas carved along the cliff are also covered by pavilion-style buildings, such as Cave 96 (North Statue) and Cave 130 (South Statue) of the Mogao Grottoes of Dunhuang, Leshan Giant Buddha, etc. Because Buddha statues were carved based on cliffs, without any back space, it was not possible to achieve horizontal encircling story-by-story during the vertical ascent process, like the situation in the pavilion of the giant statue. However, in terms of viewing form, these giant statues built in Tang are either located within a closed large cave with several open holes, or are covered by timber pavilions in an open or semi-open structure. They can be viewed through narrow stairs between the various stories of the pavilion-style building, as well as from a straight corridor outside the cave and a central lighting hole on each floor, to obtain various viewpoints in the ascending procedure story-by-story, horizontally worshiping the middle and upper parts of Buddha statues.

Specifically, the first case is the North Statue of the Mogao Grottoes of Dunhuang (Cave 96), now called the Nine-story Building. According to *Mogaoku Ji* 莫高窟記 (the records of the Mogao Grottoes) on the north wall of the front hall of Cave 15 "In the second year of the Yanzai period of the Wu Zhou Dynasty (695), Chan master Lingyin and layman Yin Zu et al. created the North Giant Statue, which was 140 *chi* high 延載二年，禪師靈隱共居士陰祖等造北大像，高一百四十尺." According to the Stele of Zhai Huaishen (831–890)[10] in the late Tang Dynasty, "I saw the North Giant Statue by the Dangquan River, which had been established for many years, but the pillars were destroyed […] The old pavilion had four eaves and cannot match the (size of) golden body (Buddha statue); after the restoration, it has five eaves, with a suitable height 乃見宕泉北大像，建立多年，棟樑摧毀……舊閣乃重飛四級，靡稱金身；新增而橫敞五層，高低得所." ([Rong 1993](), pp. 206–16). From this, it seems that the exterior of Cave 96 must have had four eaves before the restoration led by Zhang Huaishen, and after this project five eaves must have been added. The point that the previous exterior was not suitable and that the newly built (five eaves) were just suitable, is based on whether it is proportional to the height and size of the Buddha statue, although this may be an exaggerated description that belittles the old building and praises the new building. This feature reflects well the concept of "covering" in the pavilion-style architectures, which takes Buddha statues as the core of spatial design. In 1999, archaeologists discovered a platform foundation in Tang, 24.2 m wide from north to south and 9.4 m deep from east to west, and a site of a bottom hall with a width of five *jian* and a depth of two *jian* in the front of this cave ([Peng et al. 2003]()). This site is likely the bottom of the multi-story pavilion-style building in front of the grottoes of Tang.

In the photos taken by French sinologist Paul Pelliot (1878–1945) in 1908, the pavilion-style building outside Cave 96 was still in the form of a "five-story building", which had five eaves. Climbing up through narrow stairs and entering through three hole passages to the interior of the second, third, fourth and fifth story, viewers can horizontally observe the upper thighs, chest, and head of the sitting Maitreya statue inside the cave. After 1935, the pavilion outside Cave 96 was built in today's style (a nine-story building). However, from the perspective of its internal structure, there has not been a fundamental change in the viewpoints in the hole passages. Using the construction technology, such a giant statue must also be constructed in layers from top to bottom, similar to the 16th to 20th Caves of the Yungang Grottoes. Unlike the Yungang Grottoes, the viewpoints that were level with the knee, chest, chin and eyes of the Buddha statue did not disappear with the completion of the project, but were preserved through a multi-story pavilion. In Dunhuang, where there are many tourists today, it is almost impossible for the vast majority of tourists to obtain these extraordinary viewpoints through narrow stairs. But in pre modern China, it was not only possible, but also probably the intention of the designers, to observe the Buddha from different heights by climbing onto the pavilion. In addition, the Cave 130

in the Mogao Grottoes of Dunhuang, the so-called South Giant Statue, is covered by a three-story pavilion, which can also be accessed by narrow ladders. Therefore, its visual experience resembles that of the North Giant Statue, but with slightly fewer viewpoints because of the limitation of the number of stories.

From the spatial logic and visual experience of its construction, the core logic of this design concept is not fundamentally different from the Heavenly Hall, far away in Luoyang. Because of the existence of the cliff surface, the visual experience is changed from 360-degree surround viewing to the fixed viewpoint from the front of the Buddha statue. Wu Hung pointed out that the construction of this statue is another clear evidence of the dynasty's political influence on the construction of the Mogao Grottoes[…] The unusual political significance of this cave led the builders to introduce the form of the Giant Statue Cave that the Mogao Grottoes had never used before. The construction of the North Giant Statue not only added a massive landmark building to the Mogao Grottoes, but also changed their overall appearance and construction logic. Although there were many large-scale caves built before, they all belonged to the overall group of caves, and none had the dominant power of the giant statue in Cave 96. Its appearance immediately provided a powerful visual center for the overall cliff surface of the Mogao Grottoes (Wu 2022, pp. 86–88). Regarding the topic of this paper, if the Heavenly Hall enshrined with the giant statue of Maitreya in terms of height and size constituted another significant visual center beside the Bright Hall in Luoyang during the Wu Zhou Dynasty, the North Giant Statue of the Mogao Grottoes of Dunhuang (including the South Giant Statue built later) is an undeniable visual center on the multi-layer cliff surface composed of hundreds of caves. On the one hand, the towering Heavenly Hall had broken the tediously and monotonously two-dimensional layout of Luoyang; on the other hand, the vertically constructed giant statue caves and the huge pavilions in front of them break or even separate the horizontally spread grottoes and plank paths on the cliff surface. From this viewpoint, we not only discovered the unique spatial concepts and visual experiences conveyed by the two large statue caves in Dunhuang, but also discovered their inherent correlation and consistency in visual logic with the pavilion-style architecture represented by the Heavenly Hall that emerged during this period or earlier.

Another example is the Leshan Giant Buddha, which takes the visual experience offered by the pavilion-style building attached to the cliff wall to the extreme. According to *Wuchuan Lu* by Fan Chengda (1126–1193), "there is the Tianning Pavilion in the monastery, where the giant statue is located […] It has thirteen stories from the head, face to feet (of the giant statue), and is the largest Buddha statue in the world, The two ears (of the statue) are made of timbers[…] The front of the Buddha Pavilion is San'e Mountain, and the other three sides are also beautiful mountains. Multiple rivers intersect in the (canyon of) mountains. It is the first time to see the grand scene when I climb (the pavilion) since I come to *xizhou* 西州 (Nowadays Sichuan and Chongqing) 寺有天寧閣，即大像所在……爲樓十三層，自頭面以及其足，極天下佛像之大。兩耳猶以木爲之……佛閣正面三峨，餘三面皆佳山。眾江錯流諸山間。登臨之盛，自西州來始見此耳." (Fan 2012, pp. 60–61). According to the description by Wang Xiangzhi (1163–1230) in *Yudi Jisheng* 與地紀勝, "the height of the statue exceeds 360 *chi* and a seven-story pavilion is built to cover it 大像逾三百六十尺，建七層閣以覆之." (Wang 1991, p. 1038). In fact, if we consider the double-eave structure or mezzanine levels that may be used in the Pavilion of the Giant Statue, Fan Chengda's record of thirteen stories may not be fundamentally contradictory to Wang Xiangzhi's description of seven stories, either as the number of eaves or including six mezzanine levels. From the expressions of the Fan's record, such as "the two ears are made of timbers" and "the grand scene when I climb (the pavilion)", visitors must have been able to climb onto the Buddha Pavilion and view the Buddha statue closely during the Southern Song. Since the height of the Leshan Great Buddha is much greater than that of the Northern and Southern Giant Statues of Mogao Grottoes, the number of stories of this pavilion attached to the cliff is much higher than the cases in Dunhuang. Even if the mezzanine levels are not included, the number of stories of this huge pavilion was still as many as seven. Due to the

disappearance of this huge pavilion, the specific details and tangible visual experience of the building are no longer known. However, it can be imagined that the existence of such a pavilion of giant statues makes the visual experience of the huge Leshan Giant Buddha to the viewers at that time likely quite different from today. If nowadays viewers are more impressed by the enormous size of the Buddha statue, previous viewers who could climb onto the Buddha pavilion were able to horizontally gaze at various parts of the Buddha, or stand at the foot of the statue, to feel the visual intimacy and oppression due to the constant changes in viewpoints and heights. In the procedure of climbing the stairs, as the Buddha pavilion shrank upwards, they constantly approached the giant statue, gaining an increasing sense of familiarity.

After the disappearance of the huge pavilion, we can still pass through the Nine Curved Planks Path and the Lingyun Plank Path on both sides of the statue, in clockwise direction, descending from the cliff next to the Buddha's head to the Buddha's feet near the river, and then climbing up to the other side of the Buddha's head to complete the vertical circle. The Bamiyan Buddhas, located in Afghanistan, can also provide such a vertical circular path for Buddha worship, coming close to the head of the Buddha statue through hole passages. Different from the Leshan Giant Buddha, the feet of the Bamiyan Giant Buddhas and the cliffs are separated (Liu 2021, pp. 64–69). This means that people can still follow the conventional way of circumferentially worshiping the Buddha at the bottom. To sum up, the design concepts and visual logic of grottoes of giant statues and pavilions of giant statues have strong similarity and interactivity, but they are not the same. The pavilions of giant statues also have their own characteristics.

It should be emphasized that the development of pavilions of giant statues does not necessarily mean the replacement of pagodas. The development of pagodas since Tang and Song can be roughly divided into two types: the pavilion style and the dense-eave style. Whether it is a wooden structure, brick structure, or a brick–wood mixed structure, a pavilion-style pagoda can physically separate the space on each story, making it relatively independent. In the case of Yingxian Timber Pagoda, an independent space centered around Buddhist statues can be formed on each story. However, in some extreme cases, such as the White Pagoda of Qing Prefecture in Liao, although it appears to be a typical pavilion-style pagoda, the stories remain separated, without connected stairs to climb, making it impossible to achieve a visual experience from the bottom to the top. Contrarily, the number of stories inside dense-eave pagodas cannot match the number of eaves outside, and generally, internal Buddhist statues also cannot be arranged in the upper space surrounded by dense eaves. Although the early dense-eave pagodas were accessible and even had stairs to climb (such as the Small Wild Goose Pagoda in Xi'an), there was not much internal space. Additionally, apart from the first story, it was impossible to gain a truly layered and independent religious space. The later dense-eave pagodas (represented by the dense-eave pagodas of the Liao style) were mostly inaccessible, returning to the visual logic of the early-Indian and Central Asian-style stupas. They could only display their religious characteristics through external forms, allowing believers to worship around them in a two-dimensional space. For instance, the Great Pagoda in the central capital (*Zhongjing* 中京) of Liao (located on the north bank of the Laoha River in Tianyi Town and Daming Town, Ningcheng County, Chifeng City, Inner Mongolia Autonomous Region), which ranks third in height and has the largest total volume among existing Buddhist pagodas in China, is a solid, dense-eave pagoda that cannot be accessed. In a typical dense-eave pagoda of the Liao Style, the significantly elevated first story provides four or eight larger facades, allowing them to place the Four Directions Buddha, the Eight Pagodas of Sakyamunia, and other objects, forming a symbol of the Buddhist universe and time. Unlike the visual experience provided by almost contemporaneous pavilions, the Buddha statues of this type of pagoda are of shallow relief and nearly flat, unable to obtain a fully three-dimensional representation and be surrounded and worshiped as an individual statue. However, it is possible to construct a mandala pattern that combines time and space through the overall layout of the plane. Additionally, it is mainly a single-layer mandala, which is different

from the multi-layer mandala in the Yingxian Timber Pagoda (Fu 2009, p. 518; Kim 2019, pp. 53–108). The direction of observing the Buddha statue is also the external perspective in a typical open space, rather than the internal perspective in a closed space, which cannot generate a visual experience of climbing onto it and becoming close to the Buddha statue. From this perspective, the planar-surround visual logic of Buddhist pagodas after Tang maintained a considerable stability, and even greatly weakened the climbing function and internal layered sacred space of most pagodas, returning to the tradition of single-layer-surround visual logic that existed earlier, even in India.

In very few cases, people have obtained the same viewing angle as drones today through the cross-story space inside pagodas or other pagoda-style buildings, allowing them to view the base, waist, and top of the pagoda from different heights. The first case is the Feiying Pagoda in Huzhou at Zhejiang Province, known as the "pagoda in the pagoda". This pagoda is composed of stone and timber pagodas that are nested inside and outside (See Figure 7). The inner stone pagoda is over ten meters high, built in the early Southern Song. The outer timber pagoda has a multi-layer pillar network structure, built around 1234, which not only accommodates the inner pagoda but also constructs stairs that can be climbed. Due to the existence of its internal corridor, visitors can climb up the stairs and obtain a visual experience to horizontally observe the base, body, and top of the inner pagoda, even overlooking its top. Because of the core pillar placed in the center of the upper three stories of the outer pagoda, the spanning-story space inside the pagoda does not reach the three stories above the outer pagoda, but ends at the fourth story. This characteristic is the difference between the central space of the pagoda and most pavilions of giant Buddhist statues. The second case is the Putuo Zongcheng Monastery in Chengde, Hebei Province, built in 1771. In its main building, the Great Red Terrace (*dahongtai* 大红臺), there are two octagonal nanmu pagodas located in the eastern and western compartments of the south part called the secret scenic area, with a total height of 19 m and nine stories (see Figures 8 and 9). They directly reach the platform on the top of the podium of the Great Red Terrace through the hollow space spanning the second and third floors. Unlike the pavilions of giant statues, which almost exactly accommodate Buddha statues, the height of the Great Red Terrace here is not enough to accommodate the timber twin pagodas inside. Therefore, their protruding parts have surpassed the top platform on the south side of the Great Red Terrace, covered by two small pavilions with double eaves. The two timber pagodas enshrine 2160 gilded copper Buddha statues in total, each pagoda with 1080. From the second and third floors of the southern part of the Great Red Terrace, which are adjacent to the Buddhist pagodas, as well as the side windows of the small pavilions on the top platform of the Red Terrace, viewers can see the lower, middle, and top of the timber pagodas, as well as the details of the Buddha statues enshrined on each story. To sum up, if the case in Huzhou is a pagoda within a pagoda, the case of two timber pagodas in Putuo Zongcheng Monastery are pagodas within a pavilion. In terms of visual experience, the two cases above have obvious similarities with the pavilions of giant Buddhist statues, except that the huge Buddha statues in the hollow of the building are replaced by relatively small Buddha pagodas. This special phenomenon can be seen as an example of the interaction between Buddhist pagodas and pavilions of giant statues in terms of visual experience. Of course, there are only a few pagodas that can provide such visual experiences, and they cannot change their basic visual logic based on stories.

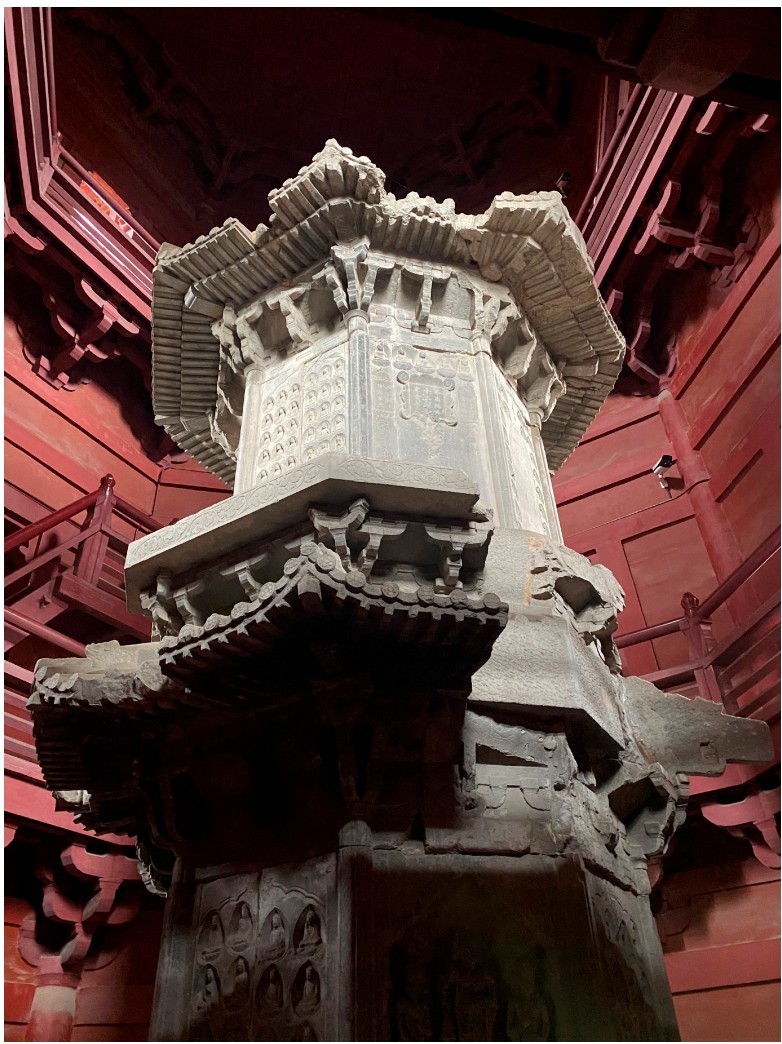

**Figure 7.** Interior view of the Feiying Pagoda (author's own photo).

The high-rise buildings of Buddhism in medieval China were represented by pagodas and pavilions, each presenting different spatial logic and visual experience. However, the similarity and interactivity between the two of them is undeniable. A scholar of Chinese architecture, Fu Xinian, believes that "the Buddha pavilions and the Buddha pagodas are both multi-story buildings from a structural perspective, and there is a certain correlation between their rise and fall. Due to the use of core pillars or an earth entity in early timber pagodas, only Buddha statues can be set up around the center structure on the ground floor. Therefore, the size and quantity of Buddha statues are limited. From the late Southern and Northern Dynasties to the Sui and Tang Dynasties, the practice of creating large statues gradually became popular, and high-rise buildings with hollow space and large statues in them obtained the key positions in Buddhist monasteries. With the development of the Buddha Pavilion, the structural style of the pagoda had also begun to change, gradually absorbing the structural characteristics of multi-story pavilions, and even resembling them in appearance." (Fu 2009, p. 511) As there are currently no surviving timber pagodas before Tang, we cannot yet determine whether early timber pagodas can only arrange Buddhist statues around the core pillar or in the central earth entity on the ground floor. However, from the cases of several central pagoda pillar caves, Cao Tiandu Pagoda, and Zhakou White Pagoda, if they are considered as models of timber pagodas of the same period, the Buddha statues are also distributed above the second story and may not be limited to the first story. Fu Xinian noticed the possible impact of the buildings of hollow multi-story pavilion style on the structure and even appearance of the timber pagoda, which is

indeed confirmed by existing buildings such as the Yingxian Timber Pagoda. From a structural perspective, due to the existence of a double-layer pillar network, it is theoretically feasible to remove the central part of the floor slabs that divide each floor, forming a hollow space to span five floors in the case of the Yingxian Timber Pagoda. The problem is that although the structure type of the double-layer pillar network of the Yingxian Pagoda is almost equivalent to the pillar network in the high pavilions that accommodated huge statues during the same period, the way to layered planar worship of Buddhas remains unchanged in these pagodas. In other words, the similarity in structure highlights the fundamental differences in spatial logic and visual experience. Fu Xinian believed that because of the resolution of structural difficulties, there was once a peak period of construction of high-rise timber pagodas from the Five Dynasties to the Liao and Song dynasties. High-rise timber pagodas, like the Sakyamuni Pagoda in the Fogong Monastery in Ying County, which adopted the structure of palace halls, were not an accidental phenomenon at that time, and must be directly related to the development of Buddhist pavilions in the Tang Dynasty (Fu 2009, p. 512). In my opinion, the emergence of pavilions of giant statues and their prevalence in Buddhist space considerably changed the spatial logic and visual experience provided by pagodas. Although the relation between pavilion and pagoda is not substitutive but parallel, because the giant statues in pavilions can offer believers a greater visual impact, and the integration of Buddha statues with architecture becomes closer, the pavilion replaced the core position of pagodas in some monasteries during the mid-to-late Tang (Fu 2009, p. 511), or shared the visual center with pagodas, occupying the axis of the entire monastery.

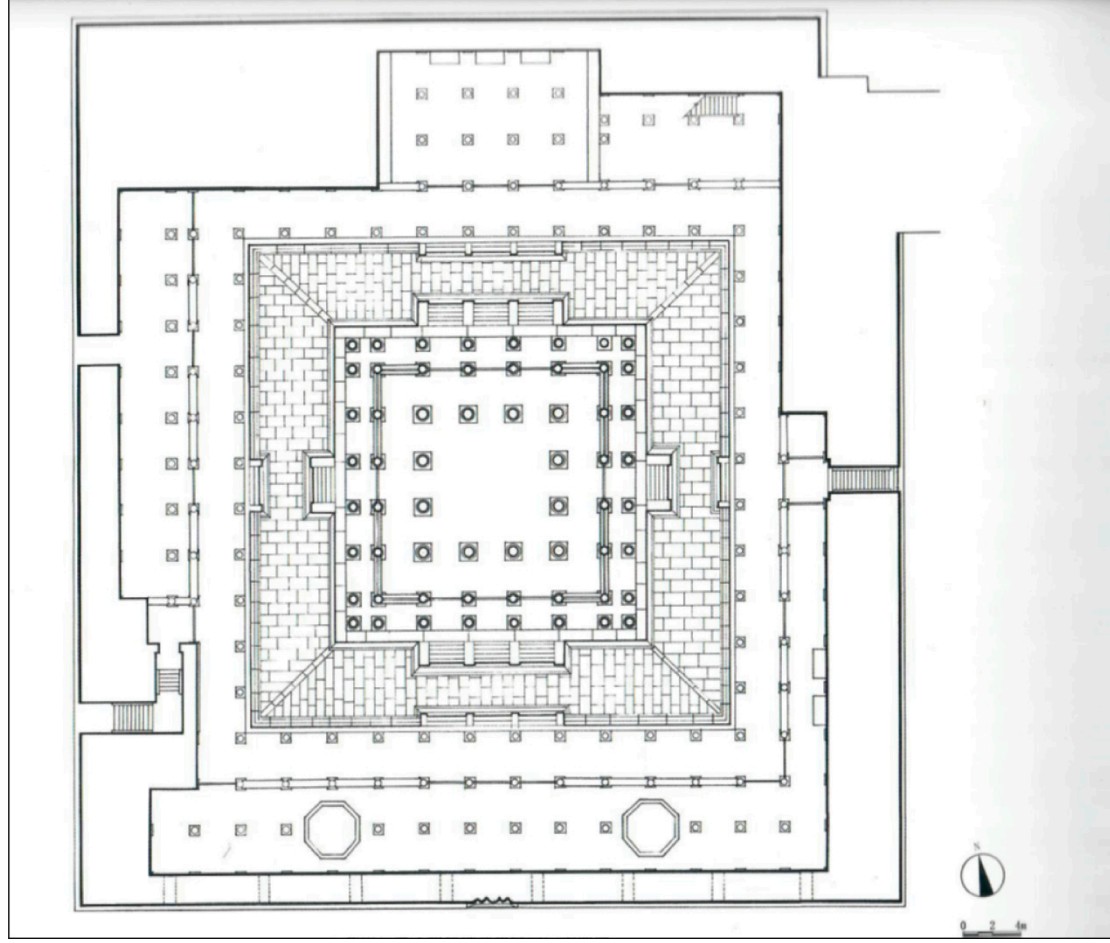

**Figure 8.** Plan of the first floor of buildings around the Dugang of Putuo Zongcheng Monastery (Yang and Zhu 2019, p. 196).

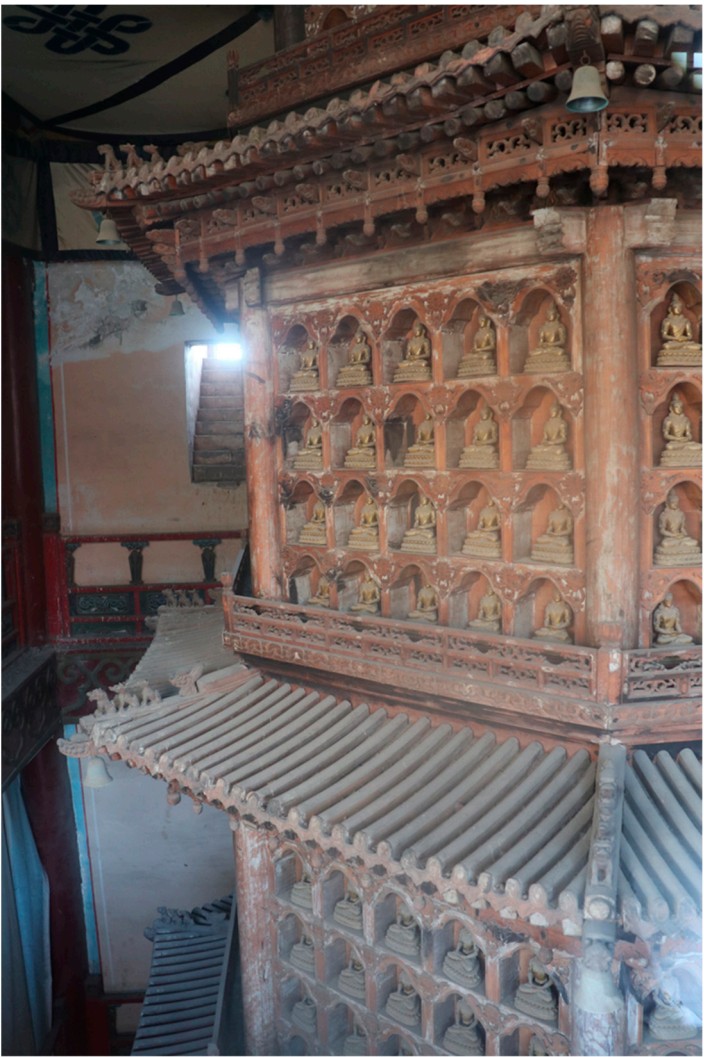

**Figure 9.** The double-timber pagoda on the southern part of the Great Red Terrace from the viewpoint of the second-floor corridor (author's own photo).

## 5. Conclusions

In summary, although the form, structure, and decorative style of stupas or pagodas and Buddhist grottoes have undergone significant transformation in the process of dissemination and development from India and Central Asia to China, their way of worship of planar detour and horizontal visual logic has not changed. Until the emergence of the pavilion, especially the development of pavilions of giant statues, viewers could not only look up at the Buddha and worship in parallel circles, but also gained a three-dimensional perspective—horizontally observing the Buddha's shoulders, neck, and head. The relationship between Buddha statues and architecture has also changed, becoming centered on Buddhist statues rather than centered on architectural structure. In Buddhist grottoes, the early distribution of single-layer or multi-layer caves horizontally on the cliff was also changed due to the excavation of the caves of giant statues spanning multi-layers, resulting in a visual experience like the pavilion of the large statue, but its viewing angle was limited to the front of the Buddha statue. The transition of spatial logic and visual experience is undoubtedly influenced by technological innovation, and is the result of development of interpretations and understandings on the Buddhism worship. The new spatial logic and visual experience, on the one hand, make the Buddha statue even higher; it can only be viewed from the bottom of the building, and it difficult to see the full view. On the other hand, viewers who climb onto it can look horizontally at the Buddha statue, to observe

its details, and even touch the fingertips of the giant statue, developing into a privilege for some believers. During this process, structural changes reshaped the logic of space. The tension between holiness and closeness, which is also bi-directional, increased in this creative visual experience, resulting in a dramatic fusion. It is precisely because of the existence of such pavilions of giant statues that the integration between the architectural structure and the sacred subject is closer, and the visual experience and spatial experience of the viewer are more abundant, compared with pagodas; the powerful religious sacredness and the approachable religious care have become possible at the same time.

The phenomenon that does not match our general common sense is the following: the towering and majestic pagodas are essentially dominated by a two-dimensional plane in terms of their religious spatial logic and visual experience, both externally and internally; however, the pavilion, which has relatively limited external height and visual impact (usually only two to three stories, up to five stories at most in the Heavenly Hall), has brought unprecedented shock to the viewer, due to its interconnectedly internal space and corresponding Buddha statue presentation, making it possible for the truly three-dimensional religious space and visual experience to constantly change with the viewer's position. From pagoda to pavilion, it is not only a transformation of the spatial logic and visual experience of high-rise Buddhist buildings for worship in medieval China, but also a visual expression of the concepts and characteristics of Sinicized Buddhism.

**Funding:** This research received no external funding.

**Data Availability Statement:** Data are contained within the article.

**Acknowledgments:** The draft of this paper was presented in the workshop of Young Scholars on the topic of "East Asia Research in the Middle Period" held at Shandong University from July 1 to 3 in 2023, and I received comments and suggestions from Sun Qi and Fang Yuan at al. And I also should thank Qin Zijin, my master student at Yuelu Academy of Hunan University, helped with the figure improvement of this article. During the procedure of paper revision, I received proofreading from Xu Caiyang of Stanford University and many suggestions from two anonymous reviewers and Wang Yudong of the University of Macau.

**Conflicts of Interest:** The author declares no conflict of interest.

## Notes

[1]  There is considerable abundant research on Chinese pagodas, and here I will only list some classic and systematic studies. See (Sirén 1930; Boerschmann 1931; Murata 1986; Zhang 2006). On the systematic research of architectural history related to pagodas in medieval China, see (Liang 1984; Steinhardt 1984, 1997, 2017, 2019).

[2]  On the earliest systematic research on the Guanyin Pavilion, see (Liang 2001, pp. 161–223). Original version published in 1932. For more details and pictures on this pavilion, see (Yang 2007). On the typical research of plane layout and space art of the Dule Monastery, especially the Guanyin Pavilion, see (Cao 1984, pp. 30–41; Zhang 1984, pp. 42–46).

[3]  The period started from the end of the Eastern Han (25–220) to a stage of what is labeled a transition during the Tang–Song periods, as defined by Naito Konan and Miyazaki Ichisada. See (Liu 1992, pp. 10–18, 153–241).

[4]  On the archaeological reports on these pagoda sites during the Northern and Southern Dynasties period, see (Datongshi Bowuguan 2007, pp. 4–26; Zhang et al. 1992, pp. 29–37, 59; Liaoningsheng Wenwu Kaogu Yanjiusuo, and Chaoyangshi Beita Bowuguan 2007; Zhongguo Shehui Kexueyuan Kaogu Yanjiusuo Luoyang Gongzuodui 1981, pp. 223–24; Zhongguo Shehui Kexueyuan Kaogu Yanjiusuo 1996; Zhongguo Shehui Kexueyuan Kaogu Yanjiusuo, and Hebeisheng Wenwu Yanjiusuo Yecheng Kaogudui 2010, pp. 31–42; Zhongguo Shehui Kexueyuan Kaogu Yanjiusuo, and Hebeisheng Wenwu Yanjiusuo Yecheng Kaogudui 2016, pp. 563–91).

[5]  *jian* is the basic unit in Chinese traditional architecture. *jian* is the space surrounded by four pillars.

[6]  On the related photos, see "Bainianqian de Hebei Zhengding", https://baijiahao.baidu.com/s?id=1762586890977683859 (accessed on 14 March 2024).

[7]  The item of "Mingtang" in *Tongdian* is basically the same as this. See (Du 1984, p. 1228).

[8]  Its height recorded in *Tongdian* is 294 *chi*. See (Du 1984, p. 1228).

[9]  It is absolutely not the only trend in the spatial structure and visual representation of Buddhist grottos. After the grottoes entered China, another trend in transformation that began during the Northern Wei Dynasty was the dilution of spatiality, emphasizing the visibility of Buddha and Bodhisattva statues on the surface of cliffs, and the openness of cliff statues, thus essentially reflecting

a landscape-oriented tendency of grotto statues. However, in the vast majority of cases, this visual landscape does not exist independently, but rather depends on related Buddhist monasteries. See (Li 2023, pp. 4–31).

[10] The official full title of this stele is "Chi Hexi Jiedu Bingbu Shangshu Zhanggong Dezheng zhi Bei". On its original texts, see S.3329 + S.11564 + P.2762 + S.6161 + S.6973 in Dunhuang manuscripts.

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
