# Peer review of "From Pagoda to Pavilion: The Transition of Spatial Logic and Visual Experience of Multi-Story Buddhist Buildings in Medieval China"

_religions, doi:10.3390/rel15030371_

Round 1

Reviewer 1 Report

Comments and Suggestions for Authors

This paper is a study of multi-story Buddhist buildings, pagoda and lou (tower) /ge or pavilion in medieval China from the perspective of visual experience. Such perspective is innovative. The overall study deserves academic attention for publication.

However, the paper also needs substantial improvement in writing to become a standard academic paper.

1. The author shall define “pavilion,” clearly in the very beginning. Using the English word can be good in communicating with non-academic audients. However, with the word “pavilion,” I didn’t understand what the author was talking about until I encountered the Chinese character ge on the second page. The author shall introduce ge and lou in the first place, and define them shape and characteristics, which the author didn’t do in this paper.

2. The writing of this paper is overall repetitive and wordy. The author often dances around instead of hitting the point directly.

For example, in the introduction, the author even gave a footnote to say what types of pavilions are not studied without explaining what kind of pavilion is studied in the paper.

Also in the introduction, the author said enough what the paper doesn’t do before he/she points out what the paper studies in the very end.

Another example, since it is already known in the field, the visual logic of stupa in India and Central Asia in the first a couple of paragraphs of section 1 can be synthesized into a few lines.

Italic book title in lines 927, 932, and 946.

Comments on the Quality of English Language

In general, the gramma is fine.

sometimes, the sentence is a little bit too long, with many commas. This is a typical Chinese sentence structure, not very English.  

vague pronoun reference: some of the pronouns are not clear in the paper. a following pronoun often refers to the Subject of the previous sentence. Each pronoun should have a clear and unmistakable noun antecedent. E.g. “it” in Line 58, is better replaced with “this paper/study.”

 If there is an original Chinese term, I would encourage the author to provide the original Chinese word and characters, e.g., “Heaven Hall.”

Author Response

To the reviewer:

   Thanks for your evaluation and suggestions for my paper. Here I will interpret my revisions in this paper to respond your comments and suggestions.

  1. For this problem, I offered the Chinese word, ge閣 in the abstract of my new version. And then in the first paragraph, after the introductions on stupas in India and Central Asia, and pagodas in China, I offered a definition on pavilion and limited the range of this concept. I said: “Therefore, we can take the pavilion as the most multi-story Buddhist building with Chinese characteristics. The pavilions defined here are limited to Buddhist space, not include later more generalized pavilions, such as library pavilions, small pavilions in Chinese gardens, or pavilions on top of large buildings.”
  2. I removed the footnote to the main text and highlight “the pavilions defined here are limited to Buddhist space”. The reviewer said: “Also in the introduction, the author said enough what the paper doesn’t do before he/she points out what the paper studies in the very end.” Because there are too many previous researches about Chinese pagodas and some researches about Chinese pavilions, I wish to highlight my main concern in this research is basically different with the previous studies, to focus on the visual experience rather than styles, structures, and decorations. The reviewer said: “since it is already known in the field, the visual logic of stupa in India and Central Asia in the first a couple of paragraphs of section 1 can be synthesized into a few lines.” I re-evaluated these two paragraphs and then I think that the first paragraph is still necessary to guide readers to know the know the basic visual logic of stupa in India and Central Asia. It is very significant to help us to understand its tiny differences and rough continuation to Chinese pagodas. But like the reviewer said, too many details and historical records are not necessary. For this reason, I deleted the whole body of paragraph 2.
  3. I Italicized the book titles in lines 927 and 932. But the title in the line 946 is not a book, but an archaeological report (in the other words, it is a paper). For this reason, I kept the original format.
  4. For the quality of English language, the reviewer said: “sometimes, the sentence is a little bit too long, with many commas. This is a typical Chinese sentence structure, not very English.” Frankly, it is hard to solve in short time. However, in some cases, I still try to split some difficult to understand long sentences with many commas into two short sentences. For some of the pronouns are not clear in the paper, I also checked my paper and made some revisions. For instance, I used “this paper” to replace “it” in Line 58.
  5. Finally, I provide the original Chinese words and characters for all Chinese terms (such as the Heaven Hall天堂, ludou櫨斗 and yongding Pillars永定柱) and Chinese resources (such as Jiutang Shu舊唐書 and Zizhi Tongjian資治通鑒).

Many thanks and best regards.

                                                          Author

Reviewer 2 Report

Comments and Suggestions for Authors

The topic is interesting. As the author stated, the paper is not focused on technical or structural comparisons of the two, pagoda and pavilion, and not focused on style analysis from an art history perspective. Although there are some conjectures at times due to lacks of substantial textual evidence and extant examples, it makes sense that Buddhist pavilions of great statue contributed to a new visual experience in the monasteries.

On Line 119-122 the author wrote "In ancient China, which lacked remarkably high buildings, the visual experience brought by such multi-story buildings was very attractive and impactful, and was also related to the power of the privileged group due to their rarity." Something that can be discussed here is that multi-storied timber buildings had already existed during the Han period, in the form of watchtowers, as evidenced by plenty of archaeological findings. This happened before the Northern and Southern Dynasties, the historical time that the author discusses as a critical time for "groundbreaking" change of visual experience in pagodas with corridors. How the popularity of the Han-period watchtowers and this "groundbreaking" change during Northern and Southern Dynasties are interrelated needs to be addressed in the paper. 

Some suggestions:

1) Line 464-469, the author discusses "the so-called yong 顒" and "the yi section" and the "yi" is critical to the argumentation here. But the author provides the character for "yong", yet omits the character for "yi", which also makes it hard for readers to follow. The "yi" character should be shown here. 

2) Figure 8 should show elevation of the structure since the author is discussing visual space in the building. 

3) In several places, necessary information of "relevant historical records" is missing. For instance, Line 402-403: "According to relevant historical records, the nature of so-called Heaven Hall was the Buddha Hall, which accommodated a giant statue of Maitreya." However, the author does not explain what exactly the "relevant historical records" are as mentioned. Also, Line 422-423, "However, in my opinion, if this site is indeed the site of the Heaven Hall, there is indeed a giant Maitreya Buddha statue that spans multiple stories in it according to historical records." Again, the author does not provide the exact "historical records" as mentioned. This kind of problem occurs again later, for instance, Line 434-435, "If the size of this Maitreya statue was indeed as huge as described in historical records, it was highly likely...". No information of the "historical records" mentioned here. 

4) Sometimes necessary explanations of certain technical terms are missing: Such as Line 546 "ludou", Line 548-549 and 554 "yongding pillars", no explanation of what they are architecturally, which makes it impossible for some readers to follow the discussions. 

5) Throughout this paper, when the author cites historical texts, there are no English translations of the titles and no historical dates of them. For instance, Line 95 cites "Foguo ji", Line 194 cites "Weishu shilao ji", Line 296 cites "Shuntianfu zhi" and "Xijin zhi", Line 493-494 cites "chaoye Qianzai", no English translation of the meaning and no information of the historical times the book was written. This makes it unclear to the readers and hard to follow the discussion since the author is trying to show the development process of pagoda or grottoes in different historical times. 

6) The author should try to avoid reliance on second-hand modern studies as historical records. For example, Line 455-458 "According to Sui Tang Jiahua, “During the reign of Emperor Zhongzong, in order to fulfill the wishes of Empress Wu, he cut off the Buddha (Maitreya) statue, shortened it, and established a new pavilion in the Shengshan Monastery to accommodate it.” xii (Liu 1979, p. 38)". The author should try to read the original historical records instead of relying on second-hand materials. 

7) Line 785-786 mentions "the Great Picture Pavilion" but this building is not discussed in anywhere of the paper.

Comments on the Quality of English Language

The English presentation of this paper needs to be much improved, as shown below:

1) Inaccurate translations of historical texts: For instance, the author tens to say "so-called" and "fundamentally" so frequently, repeatedly nearly everywhere, and although this can be considered as the author's personal writing style, it is misleading when the author even uses "so-called" in translating historical texts that not necessarily contain that meaning or term. For instance, Line 463-464, Li Chuo's text, "From the top to the so-called yong...The hole in the forehead (so-called baihao) can accommodate items weighing eight shi". Also, Line 671-677, Fan Chengda's text, "...so-called xizhou". Such translations altered the original historical texts and is misleading to readers, which should be avoided.  

2) Throughout the paper there are some levels of repetitions. For instance, Line 785-788 "This special phenomenon can be seen as an example of the interaction between Buddhist pagodas and the Great Picture Pavilion in terms of visual experience. This special phenomenon can be seen as an example of the interaction between Buddhist pagodas and the pavilion of giant statues in terms of visual experience."

The paper could be made much shorter after redundant sentences are cut. 

3) The author has an extensive list of references in the end, but some of them are not cited in the paper, for instance: Steinhardt, 1984, 1997, 2017, 2019. It is unclear what the role those uncited reference materials played in this paper.  

4) There are many places containing syntax errors and typos, sometimes making the sentences unclear. Just some examples:

Line 156-159: "Although there are a few cases of statues in the existing large Buddhist halls had already exceeded the limit of one zhang and six chi. due to the limitations of a single-story structure, Buddhist monasteries generally still cannot accommodate giant statues of tens of meters high."

Line 238-239: "Generally, the key structure that can provide this visual experience is similar the cross story hollow structure that exists in today’s large shopping malls." Later, Line 349-350 "it is similar the record of “the pavilion is thirteen zhang high” in the historical resources." Later, Line 820-822 "The problem is that although the structure type of double layer pillars network is quite similar the high pavilions that worshipped huge statues during the same period".

Line 247-248 "due to a large number of imperial and aristocratic residences were donated to Buddhism as monasteries".

Line 268-269 "there was Maitreya Pavilion with high of three stories and wide of seven jian".

Line 290-294 "Based on the height of the fragments, it can be inferred that the height of cast iron statue, which was originally about 7 meters long, and it was later destroyed during the extermination of Buddhism in the period of Huichang (841-846) (Zheng 2006, pp. 206-14; Zheng 2022, 293 pp. 30-36)."

Line 308-310 "On the one hand, its two-story design and basic spatial composition created by the Guanyin statue inside, and even the lighting setting of the two-story open windows, almost like the Yungang Caves 16 to 20." (incomplete sentence)

Line 314-315 "Instead, it has become a part of its design philosophy and visual preset." (?)

Line 316-317 "The other typical case if the Dabei Pavilion of Longxing Moastery in Zhengding County, Hebei Province." ("if" should be "is"?)

Line 415-416 "the viewing positions at different heights in heaven can reach eight different layers". ("in heaven" or in Heaven Hall?)

Line 428 "the height of statue was not match the design height of the pavilion."

Line 453 "Later, heaven was burned down" (again, "heaven"? Heaven Hall?)

Line 493 "height of heaven" (again, "heaven"? Heaven Hall)

Line 502-504 "We often see the words fu (cover) and rong (accommodate) in historical records related to the pavilion of giant Buddhist statues. This term emphasizes the inherent centrism of Buddha statues in the structural logic of Pavilions of Giant Buddha Statue..." (2 words are mentioned here, "fu" and "rong", so it is unclear which term the author refers to for "This term". Also, "centrism" may sound fancy, but is hard to understand in the context.) 

Line 714-715 "it can be roughly divided into two types: the pavilion-style style and the dense-eave style."

Line 840-844 "In summary, to follow the tradition of the stupas and early Buddhist grottoes originating from India to Central Asia, although their form, structure, and decorative style of Chinese pagodas have undergone significant changes in the development process over hundreds of years; in terms of their planar detour concept and visual logic, it can be said that they have achieved vertical multi-layer stacking, but have not undergone fundamental changes." (unclear sentence, it needs to be rewritten.)

The paper also contains some mistakes in Chinese pinyin spellings.

Author Response

To the reviewer:

   Thanks for your evaluation and suggestions for my paper. Here I will interpret my revisions in this paper to respond your comments and suggestions.

  • About the popularity of the Han-period watchtowers, I rewrite this paragraph to respond this point. “In ancient China, although that multi-storied timber buildings had already existed during Han Dynasty (BCE202- CE220), in the form of watchtowers, as evidenced by plenty of archaeological findings, the visual experience brought by such multi-story buildings was limited in personal space and particular groups like sentinels. For rulers, officials and religious group, this visual experience was still very attractive and impactful, and was also related to the power of the privileged group due to their rarity during the Northern and Southern Dynasties (420-589).”
  • Line 464-469, I have already shown “yi” character 頤
  • In the Figure 8, we do not have the data of elevation of the structure. The way to solve this problem in my new version is to add Figure 9, a photo taken by myself, to show the spatial relationship between the internal timber pagodas and the external building.
  • The reviewer pointed our that in several places, necessary information of "relevant historical records" is missing, such as Line 402-403, 422-423 and 434-435. I have already added information of these historical records in these three cases and checked other cases.
  • I have already added necessary explanations of certain technical terms, such as “ludou” and “yongding pillars”.
  • I have already offered the Chinese characters for these historical materials, such as Foguo Ji and Weishu Shilao Zhi. At the same time, I also try to offer the English translation for the titles of these materials or pointed out the historical dates of them.
  • In the case from Line455-458. The original Chinese text is”至中宗欲成武后志,乃斫像令短,建聖善寺閣以居之。” I try to translate this sentence into “During the reign of Emperor Zhongzong, in order to fulfill the wishes of Empress Wu, he cut off the Buddha (Maitreya) statue, shortened it, and established a new pavilion in the Shengshan Monastery to accommodate it.” It is not the second-hand materials, but the original historical records by Liu Su in Suitang Jiahua, the punctuation version of this book published in 1979.
  • “The Great Picture Pavilion” is a wrong description in Line 785-786. The correct one is “pavilions of giant Buddhist statues”.
  • I checked the full text of my paper and found 20 “so-called”. And then I reassessed the necessity to use it one by one and deleted most of “so-called”. In some cases, to use “so-called” is still necessary. For instance, when we talked about the “original caves”, I try to use “so-called” to point out the other name of these three caves, “three caves in Northern Liang Kingdom [397-460]”, because some scholars did not believe the dates of these three caves is Northern Liang.
  • I have already deleted the obvious repetitions in Line 785-788 and checked other parts of my paper.
  • I cited Professor Steinhardt’s works in 1984, 1997, 2017 and 2019 in the second note of my paper. “On the systematic researches of architectural history related to pagodas in Medieval China, see (Liang 1984; Steinhart 1984, 1997, 2017 and 2019).”
  • Line 156-159: “Although there are a few cases of statues in the existing large Buddhist halls had already exceeded the limit of one zhang and six chi. due to the limitations of a single-story structure, Buddhist monasteries generally still cannot accommodate giant statues of tens of meters high." The revised sentences are “Although there are also a few statues in the existing Great Buddha Hall that exceed the height limit of one zhang and six chi. Due to the limitations of a single-story structure, Buddhist monasteries generally still cannot accommodate giant statues of tens of meters high.”
  • For the Line 238-239: "Generally, the key structure that can provide this visual experience is similar the cross story hollow structure that exists in today’s large shopping malls." The revised sentence is “Generally, the key structure that can provide this visual experience is comparable the cross story hollow structure that exists in today’s large shopping malls. Later, the Line 349-350 "it is similar the record of “the pavilion is thirteen zhang high” in the historical resources." The revised sentence is “it roughly matches the record of ‘the pavilion is thirteen zhang high’ in the historical resources”. Later, the Line 820-822 "The problem is that although the structure type of double layer pillars network is quite similar the high pavilions that worshipped huge statues during the same period". The revised sentence is “The problem is that although the structure type of double layer pillars network of Yingxian Pagoda is almost equivalent to pillars networks in the high pavilions that accommodated huge statues during the same period”.
  • Line 247-248 "due to a large number of imperial and aristocratic residences were donated to Buddhism as monasteries". The full revised sentence is “On the one hand, there are similarities of special structure between Buddhist monasteries and imperial or aristocratic high-ranking buildings, due to many imperial and aristocratic residences were donated to Buddhism as monasteries at that time, and on the other hand, it was also related to the way of setting up of Buddhist statues.”
  • Line 268-269 "there was Maitreya Pavilion with high of three stories and wide of seven jian". The revised sentence is “there was Maitreya Pavilion of three stories and a width of seven jian間”.
  • Line 290-294 “Based on the height of the fragments, it can be inferred that the height of cast iron statue, which was originally about 7 meters long, and it was later destroyed during the extermination of Buddhism in the period of Huichang (841-846) (Zheng 2006, pp. 206-14; Zheng 2022, 293 pp. 30-36)." The revised sentences are: “Based on the height of the fragments of the cast iron statue, it can be inferred that its original height was about 7 meters. The originally complete statue was later destroyed during the extermination of Buddhism in the period of Huichang (841-846) (Zheng 2006, pp. 206-14; Zheng 2022, pp. 30-36).”
  • Line 308-310, the complete sentence is “On the one hand, it is comparable to the Yungang Caves 16 to 20 that the two-story design and basic spatial composition created by the Guanyin statue inside, and even the lighting setting of the two-story open windows in the Guanyin Pavilion.”
  • Line 314-315, the “visual preset” should be change to “visual logic”.
  • Line 316-317, "if" should be "is".
  • Line 415-416, “in heaven” should be “in the Heaven Hall”.
  • Line 428, “was not match” should be “was not matched”.
  • Line 453, “heaven” should be “the Heaven Hall”.
  • Line 493, “heaven” should be “the Heaven Hall”.
  • Line 502-504, “This term emphasizes the inherent centrism of Buddha statues in the structural logic of Pavilions of Giant Buddha Statue...” The revised sentence is “These Chinese words like fu and rong emphasize the core position of Buddha statues in the structural logic of Pavilions of Giant Buddha Statue, and explains the primary and secondary relationships between Buddha statues and pavilions.
  • Line 714-715, “the pavilion-style style” should be “the pavilion-style”.
  • Line 840-844, the revision of this unclear sentence is: “In summary, although the form, structure, and decorative style of stupas or pagodas and Buddhist grottoes have undergone significant transformation in the process of dissemination and development from India, Central Asia to China, their worship way of planar detour and horizontal visual logic did not change basically.”
  • I also checked the Chinese pinyin spellings in the full text and correct some mistakes.

Author